# Explaining Caption-Image Interactions in CLIP Models with Second-Order Attributions

**Lucas Moeller**[*]                     *lucas.moeller@ims.uni-stuttgart.de*
**Pascal Tilli**[*]                         *pascal.tilli@ims.uni-stuttgart.de*
**Ngoc Thang Vu**                                 *vu@ims.uni-stuttgart.de*
**Sebastian Pado**                             *pado@ims.uni-stuttgart.de*
*University of Stuttgart*

**Reviewed on OpenReview:** *https://openreview.net/forum?id=HUUL19U7HP*

## Abstract

Dual encoder architectures like CLIP models map two types of inputs into a shared embedding space and predict similarities between them. Despite their wide application, it is, however, not understood *how* these models compare their two inputs. Common first-order feature-attribution methods explain importances of individual features and can, thus, only provide limited insights into dual encoders, whose predictions depend on interactions between features.

In this paper, we first derive a second-order method enabling the attribution of predictions by any differentiable dual encoder onto feature-interactions between its inputs. Second, we apply our method to CLIP models and show that they learn fine-grained correspondences between parts of captions and regions in images. They match objects across input modes and also account for mismatches. This intrinsic visual-linguistic grounding ability, however, varies heavily between object classes, exhibits pronounced out-of-domain effects and we can identify individual errors as well as systematic failure categories. Code is publicly available: https://github.com/lucasmllr/exCLIP

## 1 Introduction

Dual encoder models use independent modules to represent two types of inputs in a common embedding space and are optimized to predict a scalar similarity measure for them. The training objective is typically a triplet or contrastive loss (Sohn, 2016; van den Oord et al., 2019). Popular examples include Siamese transformers for text-text pairs (SBERT) (Reimers & Gurevych, 2019) and Contrastive Language-Image Pre-Training (CLIP) models (Radford et al., 2021; Jia et al., 2021) for text-image pairs. The learned representations have proven to be highly informative for downstream applications, such as image classification (Zhang et al., 2022a), visual question answering (Antol et al., 2015; Tilli & Vu, 2025), image captioning and visual entailment (Shen et al., 2021), as well as text or image generation (Chen et al., 2023a; Yu et al., 2022; Rombach et al., 2022). In (multi-modal) information retrieval, dual encoders enable efficient semantic search (Baldrati et al., 2022; Zhu et al., 2024; Formal et al., 2021; Xiong et al., 2021; Johnson et al., 2019), serving e.g. Retrieval-Augmented Generation (RAG) Gao et al. (2023)

Despite the wide-spread application of dual encoder models, an open question remains *how* these models compare their two inputs. Common first-order attribution methods like Shapley values (Lundberg & Lee, 2017) or Integrated Gradients (IG) (Sundararajan et al., 2017) can only provide limited insights into dual encoders because they attribute to *individual* features (Zheng et al., 2020; Ramamurthy et al., 2022; Janizek et al., 2021; Sundararajan et al., 2020). However, similarity fundamentally depends on comparisons and, therefore, on *interactions* of features (Tversky, 1977; Lin, 1998). In dual encoders this manifests in the final

---

[*]These authors contributed equally.

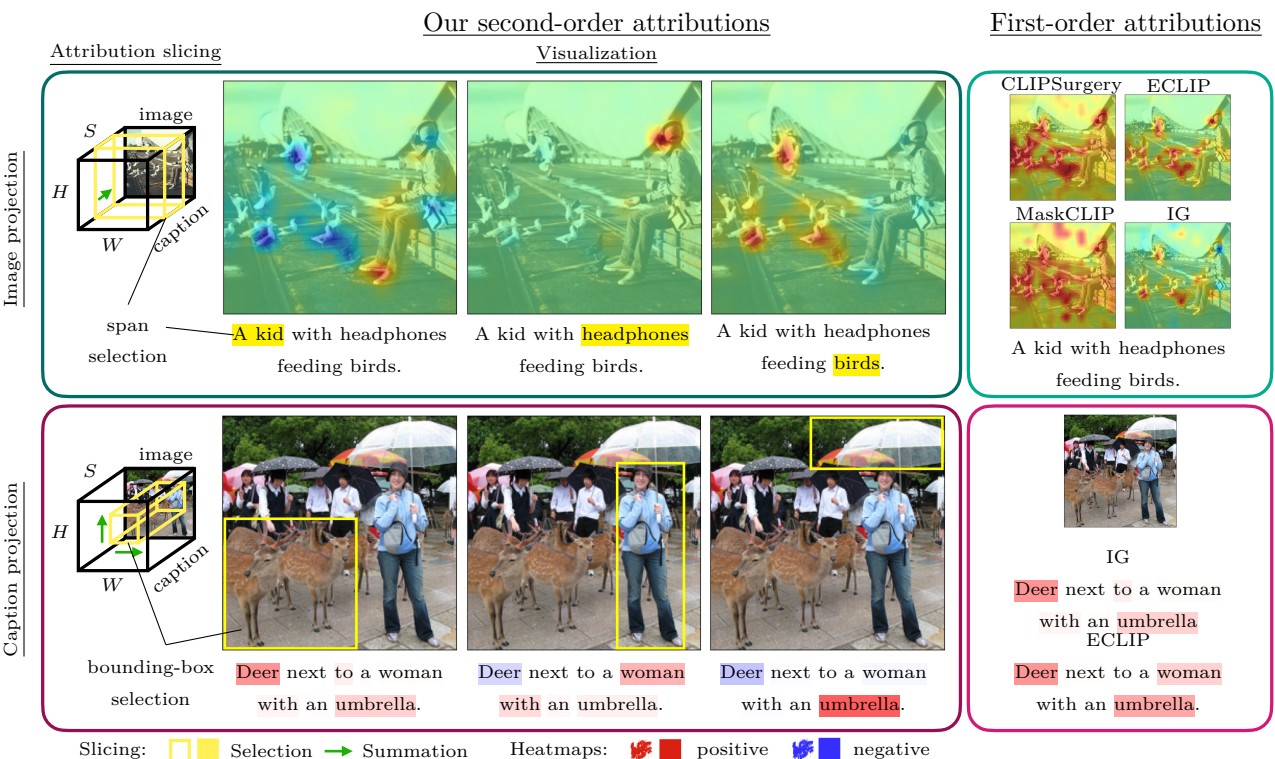

Figure 1: (Left column) **Our second-order attributions can point out *interactions* between arbitrary spans in captions and regions in images**. We can visualize them by slicing (yellow selection) our 3d attribution tensor with image dimensions $(H, W)$ and caption dimension $S$ (details in Section 3). A selection can be projected onto the image (top-left) or the caption (bottom-left) by summation (green arrows). Heatmaps for these projected attributions are in shades of red/blue for positive/negative values. (Right column) In contrast, first-order attributions can only attribute the overall similarity between captions and images onto either the image (top-right) *or* the caption (bottom-right). They *cannot* assess underlying interactions.

cosine-similarity of the two embeddings, resulting in all terms contributing to the output similarity score to contain multiplicative dependencies between the two inputs. In such multiplicative terms, a change in one involved feature affects the contribution of others; hence, these features interact.

Only few works have studied feature interaction in symmetric Siamese encoders (Eberle et al., 2020; Möller et al., 2023; 2024; Vasileiou & Eberle, 2024) and they have remained almost entirely unstudied in non-symmetric dual encoders like CLIP (Joukovsky et al., 2023).

In this work, we address this research gap and aim at a means to analyze which aspects in two given inputs dual encoders compare in order to predict a similarity for them. Our contributions are the following:

(1) Motivated by the theory behind IG (cf. Appendix F), we derive a general second-order feature attribution method that can explain *interactions* between inputs of any differentiable dual encoder model. The method does not rely on any modification of the trained model, nor on additional optimization. (2) We apply our method to a range of CLIP models and demonstrate that they can capture fine-grained interactions between corresponding parts of captions and regions in images. They identify matching objects across the input modes and also penalize mismatches. Using image-captioning datasets with object bounding-box annotations, we evaluate the extent and limitations of this *intrinsic visual-linguistic grounding ability* in a wide range of CLIP models.

Figure 1 illustrates our interaction attributions showing how they can point out corresponding parts of captions and regions in images. In contrast, first-order alternatives cannot access these interactions but only provide insights into aspects important to the *overall* similarity between a text and image input.

## 2   Related work

**Metric learning**   refers to the task of producing embeddings reflecting the similarity between inputs (Kaya & Bilge, 2019). Applications include face identification (Guillaumin et al., 2009; Wojke & Bewley, 2018) and image retrieval (Zhai & Wu, 2018; Gao et al., 2014). Siamese networks with cosine similarity of embeddings were early candidates (Chen & He, 2021). The triplet-loss (Hoffer & Ailon, 2015) involving negative examples has been proposed as an improvement but requires sampling strategies for the large number of possible triplets (Roth et al., 2020). Qian et al. (2019) have shown that the triplet-loss can be relaxed to a softmax variant. Sohn (2016) and van den Oord et al. (2019) have proposed the batch contrastive objective which has been applied in both unsupervised (Caron et al., 2020) and supervised representation learning (Khosla et al., 2020). It has led to highly generalizable semantic text (Reimers & Gurevych, 2019) and image embeddings (He et al., 2020) and ultimately to the Clip training paradigm Radford et al. (2021).

**Vision-language models**   process both visual and linguistic inputs. Zhang et al. (2022b) were the first to train a dual-encoder architecture with a contrastive objective on image-text data in the medical domain. Radford et al. (2021) and Jia et al. (2021) have applied this principle to web-scale image captions and alt-text data. In the following, the basic inter-modal contrastive loss has been extended by intra-modal loss terms (Goel et al., 2022; Lee et al., 2022; Yang et al., 2022a), self-supervision (Mu et al., 2022), non-contrastive objectives (Zhou et al., 2023), incorporating classification labels (Yang et al., 2022b), textual augmentation (Fan et al., 2023), a unified multi-modal encoder architecture (Mustafa et al., 2022) and retrieval augmentation (Xie et al., 2023). Next to more advanced training objectives, other works have identified the training data distribution to be crucial for performance: Gadre et al. (2023) have proposed the DataComp benchmark focusing on dataset curation while fixing model architecture and training procedure, Xu et al. (2024) have balanced metadata distributions and Fang et al. (2024) have introduced data filtering networks for the purpose. The strictly separated dual-encoder architecture has been extended to include cross-encoder dependencies (Li et al., 2022a; Pramanick et al., 2023), and multi-modal encoders have been combined with generative decoders (Chen et al., 2023a; Lu et al., 2023; Li et al., 2021; Koh et al., 2023; Alayrac et al., 2022; Yu et al., 2022).

**Local feature attribution methods**   aim at explaining a given prediction by assigning contributions to individual input features (Murdoch et al., 2019; Doshi-Velez & Kim, 2017; Lipton, 2018; Atanasova et al., 2020). First-order gradients can approximate a prediction's sensitivity to such features (Li et al., 2016) and gradient×input saliencies can approximate feature importance (Simonyan et al., 2014). In transformer archi-tectures, attention weights have been analyzed (Abnar & Zuidema, 2020), but were subsequently contested as explanation because they are only one component of the model (Jain & Wallace, 2019; Wiegreffe & Pin-ter, 2019; Bastings & Filippova, 2020). Layer-wise relevance propagation (Lrp) defines layer-specific rules to back-propagate attributions to individual features (Montavon et al., 2019; Bach et al., 2015). In contrast, Shapley values (Lundberg & Lee, 2017) and IG (Sundararajan et al., 2017) treat models holistically and can provide a form of theoretical guaranty for correctness. This has recently been challenged by Bilodeau et al. (2024) who proved fundamental limitations of attribution methods. A widely used attribution method in the vision domain is GradCam (Selvaraju et al., 2017), which Chefer et al. (2021) and Bousselham et al. (2024) extended to transformer architectures.
Assigning importances to individual features, first-order attribution methods cannot capture dependencies on feature interactions. Tsang et al. (2018) have proposed to detect such interactions from weight matrices in feed-forward neural networks, Cui et al. (2020) investigated them in Bayesian networks. The Shapley value has been extended to the Shapley (Taylor) Interaction Index (Grabisch & Roubens, 1999; Sundararajan et al., 2020; Fumagalli et al., 2024) and Janizek et al. (2021) have generalized IG to integrated Hessians. Plummer et al. (2020) and Zheng et al. (2020) have assessed interactions underlying similarity predictions in Siamese image encoders. Eberle et al. (2020) extended Lrp for this class of models (Vasileiou & Eberle, 2024), and our prior work extended IG to Siamese language encoders (Möller et al., 2023; 2024). In this work, we further generalize this method to multi-modal dual encoders.

**CLIP explainability.**   Several works have previously pursued the goal of better understanding Clip mod-els and contrastive image encoders. Wang et al. (2023) and Kazmierczak et al. (2024) have proposed in-

formation bottleneck approaches. Bhalla et al. (2025) identified interpretable sparse concepts in the shared embedding space. Rasekh et al. (2024) predicted human-understandable rationales for images. Quantmeyer et al. (2024) localized where the text encoder processes negation. Giulivi & Boracchi (2024) created saliency maps for WORDNET concepts. Chen et al. (2022) proposed an improved CAM variant and analyzed which objects the model looks at. Materzyńska et al. (2022) were interested in the entanglement of image representations. Gandelsman et al. (2023) identified the roles of individual attention heads in CLIP's image encoder and later investigated second-order effects of neurons (Gandelsman et al., 2025). Lewis et al. (2024) analyzed whether CLIP models adequately handle compositional concepts. Sam et al. (2024) investigated the model's reasoning ability about differences in images and Tu et al. (2024) examined safety objectives. The unseen performances of CLIP have motivated a number of authors to identify the reasons behind its ostensible generalization ability and robustness towards domain shifts (Xue et al., 2024; Nguyen et al., 2022; Fang et al., 2022; Tu et al., 2023; Mayilvahanan et al., 2024; 2025). Zhao et al. (2024) explored a wide range of first-order methods to attribute similarity scores to images and captions independently and Li et al. (2023) proposed the CLIPSURGERY method. Sammani et al. (2023) and Lerman et al. (2021) independently introduced a second-order variant of GRADCAM that can assess feature interactions. It can be applied to CLIP; in Appendix G, we show that it is a special case of our method. Most closely related to our work, INTERACTIONLIME (Joukovsky et al., 2023) pioneered the attribution of interactions between captions and images in CLIP models. However, relying on a local bilinear approximation of CLIP, it does not explain the original model and requires additional optimization as well as hyper-parameter tuning (cf. Appendix H). Last, ITSM by Li et al. (2022c) and the method by Black et al. (2022) are forward-facing saliency methods that compute importance values through pair-wise embedding multiplication. We compare these approaches against ours in Section 4.2.

**Visual-linguistic grounding** refers to the identification of fine-grained relations between text phrases and corresponding image parts (Chen et al., 2023b). Specialized models predict regions over images for a corresponding input phrase (Sadhu et al., 2019; Ye et al., 2019). This objective has been combined with contrastive caption matching (Li et al., 2022b; Datta et al., 2019), and caption generation (Yang et al., 2022c). The VOLTA model internally matches latent image-region and text-span representations (Pramanick et al., 2023). In multi-modal text generative models, grounding has been included as an additional pretraining task (Li et al., 2020; Su et al., 2019; Chen et al., 2020) and can be unlocked with visual prompt learning (Dorkenwald et al., 2024). At the intersect of grounding and explainability, Hendricks et al. (2016) have generated textual explanations for vision models and have grounded them to input images (Hendricks et al., 2018; Park et al., 2018). In this paper, we do not optimize models to explicitly ground predictions, but aim at analyzing to which extent purely contrastively trained dual encoders acquire this ability intrinsically.

## 3 Method

We first derive general second-order attributions for dual encoder predictions enabling the assessment of feature-interactions between their two inputs. In the following, we then describe their realization in transformer models, specifically CLIP.

**Derivation of second-order attributions.** We begin from the definition of a dual encoder $f$,

$$s = f(\mathbf{a}, \mathbf{b}) = \mathbf{g}(\mathbf{a})^\top \mathbf{h}(\mathbf{b}), \tag{1}$$

with two vector-valued encoders $\mathbf{g}$ and $\mathbf{h}$, respective inputs $\mathbf{a}$ and $\mathbf{b}$ and a scalar output $s$. We also define two *reference* inputs $\mathbf{r}_a$ and $\mathbf{r}_b$, whose role will be discussed later. With these definitions, we can write the following expression,

$$f(\mathbf{a}, \mathbf{b}) - f(\mathbf{r}_a, \mathbf{b}) - f(\mathbf{a}, \mathbf{r}_b) + f(\mathbf{r}_a, \mathbf{r}_b), \tag{2}$$

which will serve as a rigorous starting-point of our derivation. In the following, we first proceed by showing the equality of this initial starting-point to Eq. 10. We then reduce this equality to our final result in Eq. 11 using the approximations discussed below. At this point, we are also discussing an intuitive interpretation of the final result.

As a first step, we see $f$ as an anti-derivative and reformulate the above expression into an integral over its derivative:

$$\left[f(\mathbf{a}, \mathbf{b}) - f(\mathbf{r}_a, \mathbf{b})\right] - \left[f(\mathbf{a}, \mathbf{r}_b) - f(\mathbf{r}_a, \mathbf{r}_b)\right]$$
$$= \int_{\mathbf{r}_b}^{\mathbf{b}} \frac{\partial}{\partial \mathbf{y}_j} \left[f(\mathbf{a}, \mathbf{y}) - f(\mathbf{r}_a, \mathbf{y})\right] d\mathbf{y}_j = \int_{\mathbf{r}_b}^{\mathbf{b}} \int_{\mathbf{r}_a}^{\mathbf{a}} \frac{\partial^2}{\partial \mathbf{y}_j \partial \mathbf{x}_i} f(\mathbf{x}, \mathbf{y}) \, d\mathbf{x}_i \, d\mathbf{y}_j \tag{3}$$

Here, $\mathbf{x}$ and $\mathbf{y}$ are integration variables for the two inputs. This step can be seen as the second-order equivalent to Equation 12 in the theory behind IG (cf. Appendix F). We use component-wise notation with indices $i$ and $j$ for the input dimensions and omit sums over double indices for clarity. We then plug in the model definition from Equation 1,

$$\int_{\mathbf{r}_a}^{\mathbf{a}} \int_{\mathbf{r}_b}^{\mathbf{b}} \frac{\partial^2}{\partial \mathbf{x}_i \partial \mathbf{y}_j} \mathbf{g}_k(\mathbf{x}) \, \mathbf{h}_k(\mathbf{y}) \, d\mathbf{x}_i \, d\mathbf{y}_j, \tag{4}$$

again using component-wise notation for the dot-product with $k$ indexing the dimension of the shared embedding space. Since neither embedding depends on the other integration variable, we can separate both integrals and derivatives applying a product rule:

$$\int_{\mathbf{r}_a}^{\mathbf{a}} \frac{\partial \mathbf{g}_k(\mathbf{x})}{\partial \mathbf{x}_i} \, d\mathbf{x}_i \int_{\mathbf{r}_b}^{\mathbf{b}} \frac{\partial \mathbf{h}_k(\mathbf{y})}{\partial \mathbf{y}_j} \, d\mathbf{y}_j \tag{5}$$

Both terms are line integrals from the references to the actual inputs in the respective input representation spaces; $\partial \mathbf{g}_k(\mathbf{x})/\partial \mathbf{x}_i$ and $\partial \mathbf{h}_k(\mathbf{y})/\partial \mathbf{y}_j$ are the Jacobians of the two encoders. To proceed with these integrals, we define integration paths and substitute. We follow Sundararajan et al. (2017), and use the straight lines between both references and inputs,

$$\mathbf{x}(\alpha) = \mathbf{r}_a + \alpha(\mathbf{a} - \mathbf{r}_a), \tag{6}$$
$$\mathbf{y}(\beta) = \mathbf{r}_b + \beta(\mathbf{b} - \mathbf{r}_b), \tag{7}$$

parameterized by $\alpha$ and $\beta$, respectively. For the integral over encoder $\mathbf{g}$ substituting the path $\mathbf{x}(\alpha)$ yields an integral over the scalar integration variable $\alpha$:

$$\int_0^1 \frac{\partial \mathbf{g}_k(\mathbf{x}(\alpha))}{\partial \mathbf{x}_i} \frac{\partial \mathbf{x}_i(\alpha)}{\partial \alpha} \, d\alpha = (\mathbf{a} - \mathbf{r}_a)_i \int_0^1 \frac{\partial \mathbf{g}_k(\mathbf{x}(\alpha))}{\partial \mathbf{x}_i} \, d\alpha \tag{8}$$

Since $\partial \mathbf{x}(\alpha)/\partial \alpha = (\mathbf{a} - \mathbf{r}_a)$ is a constant w.r.t $\alpha$, we can pull it out of the integral. We then define the *integrated Jacobian* for the encoder $\mathbf{g}$,

$$\mathbf{J}_{ki}^g := \int_0^1 \frac{\partial \mathbf{g}_k(\mathbf{x}(\alpha))}{\partial \mathbf{x}_i} \, d\alpha \approx \frac{1}{N} \sum_{n=1}^N \frac{\partial \mathbf{g}_k(\mathbf{x}(\alpha_n))}{\partial \mathbf{x}_i}, \tag{9}$$

as the analogon to integrated gradients for vector-valued models. The integral over encoder $\mathbf{h}$ can be processed in the same way by substituting $\mathbf{y}(\beta)$ to obtain $\mathbf{J}_{kj}^h$. In practice, these integrals are calculated numerically by sums over $N$ steps, with $\alpha_n = n/N$. This introduces an approximation error which must, however, converge to zero for large $N$ by definition of the Riemann integral. We plug the results from Equation 8 and the definitions of the *integrated Jacobians* into Equation 5:

$$(\mathbf{a} - \mathbf{r}_a)_i \, \mathbf{J}_{ik}^g \mathbf{J}_{kj}^h \, (\mathbf{b} - \mathbf{r}_b)_j =: \sum_{ij} \mathbf{A}_{ij} \tag{10}$$

After computing the sum over the output embedding dimension $k$, this yields interaction terms, each involving a feature pair $(i, j)$ with feature $i$ from input $\mathbf{a}$ and feature $j$ from input $\mathbf{b}$. We can write the values of these terms for all feature pairs into a matrix with index $i$ on one side and $j$ on the other, which we refer to as the *attribution matrix* $\mathbf{A}_{ij}$. In the last step, we write out the omitted sum over $i$ and $j$ explicitly. Note that except for the numerical integration, the equality to Equation 2 still holds. Hence, the sum over all feature

A hot dog sitting on a table covered in confetti. Surrounded by glitter, there is a sausage in a bun.

---

A hot dog sitting on a table covered in confetti. Surrounded by glitter, there is a sausage in a bun.

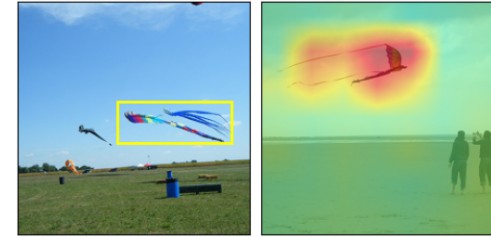

Figure 2: (Left) Intra-modal text-text attributions between top and bottom captions (top: selections in yellow, bottom: corresponding attributions in red/blue for positive/negative). (Right) Intra-modal image-image attributions between left and right image (left: bounding-box selection in yellow, right: heatmaps as above). More examples can be found in Figure 10.

pair attributions in $\mathbf{A}$ is an exact reformulation of our starting-point.

At this point, we return to the *references*, $\mathbf{r}_a$ and $\mathbf{r}_b$, defined above. We require them to be approximately dissimilar to any other input, e.g. a black image or a caption consisting of padding tokens for respective encoders. If this is the case all three terms involving $\mathbf{r}_a$ and $\mathbf{r}_b$ in Equation 2 approximately vanish, i.e. $f(\mathbf{r}_a, \mathbf{b}) \approx 0$, $f(\mathbf{a}, \mathbf{r}_b) \approx 0$, $f(\mathbf{r}_a, \mathbf{r}_b) \approx 0$. This reduces the equality between Equations 2 and 10 to our final result:

$$f(\mathbf{a}, \mathbf{b}) \approx \sum_{ij} \mathbf{A}_{ij}. \tag{11}$$

Intuitively, this provides an approximate decomposition of the model prediction $s = f(\mathbf{a}, \mathbf{b})$ into additive contributions of feature-pair interactions between the two inputs. Throughout this paper, we evaluate the attribution matrix $\mathbf{A}$.

**Interaction attributions in transformer models.** In the derivation above, we treat image and text representations as vectors. In transformer-based encoders, text inputs are represented as $S \times D_g$ dimensional tensors, where $S$ is the length of the token sequence. Image representations are of shape $H \times W \times D_h$, with $H$ and $W$ being height and width of the image representation; in vision-transformers both equal the number of patches $P$. $D_g$ and $D_h$ are the encoders' embedding dimensionalities. Our pair-wise image-text interaction attributions thus have the dimensions $H \times W \times D_h \times S \times D_g$, which quickly becomes intractably large. Fortunately, the sum over dimensions in Equation 11 enables the additive combination of attributions in $\mathbf{A}$. We sum over the embedding dimensions of both encoders $D_g$ and $D_h$ and obtain an $H \times W \times S$ dimensional attribution tensor, which estimates for *each pair of a text token and an image patch* how much their interaction contributes to the overall prediction. These attributions are still three-dimensional and thus not straightforward to visualize. However, again we can use their additivity, slice the 3d attribution tensor along text or image dimensions and project onto the remaining dimensions by summation. This projection is demonstrated in Figure 1 schematically and with examples, both for a selection over a token range in the caption (top) and a selection over a bounding-box in the image (bottom). Albeit CLIP models are typically trained to match images against captions, we can also compute intra-modal attributions for image-image or text-text pairs by applying the same encoder to both inputs. Appendix B discusses this in more detail. Figure 2 shows two examples.

## 4 Experiments

In our experiments, we apply our feature-interaction attributions to CLIP models. We focus on evaluating the interactions between mentioned objects in captions and corresponding regions in images by selecting token-ranges in captions and analyzing their interactions with image patches. In the first series of experiments, we compare our attributions against baselines (Section 4.2). The second series in Section 4.3 then utilizes our method and analyzes CLIP models.

### 4.1 Experimental setting

We base our evaluation on three image-caption datasets that additionally contain object bounding-box annotations in images, Microsoft's Common Objects in Context (Coco) (Lin et al., 2014), the Flickr30k collection (Young et al., 2014) with entity annotations (Plummer et al., 2015), and the Hard Negative Captions (Hnc) dataset by Dönmez et al. (2023). Hnc generates captions from scene graphs based on templates. We use a basic `subject predicate object` template to align with the domain of the other two datasets. We use Hnc for evaluation only, on Flickr30k we use the test split, and on Coco we use the validation split for our analysis as the test split does not contain captions[1].

We work with Clip dual encoders (Radford et al., 2021) trained with the standard inter-modal contrastive objective and analyze the original OpenAI models, as well as MetaClip (Xu et al., 2024) and the Open-Clip reimplementations trained on the Laion (Schuhmann et al., 2022), Dfn (Fang et al., 2024), Common-Pool , and DataComp (Gadre et al., 2023) datasets[2]. If not mentioned otherwise, our experiments are based on the ViT-B-16 architecture. In addition to the unmodified models, we evaluate variants fine-tuned on the Coco and Flickr30k training splits. We run all trainings for five epochs using AdamW (Loshchilov & Hutter, 2018), starting with an initial learning rate of $1 \times 10^{-7}$ that exponentially increases to $1 \times 10^{-5}$. Weight decay is set to $1 \times 10^{-4}$ and the batch size is 64 on a single 50GB Nvidia A6000.

### 4.2 Attribution evaluation

In the first series of experiments, we compare our attributions against baselines. Figure 1 includes a qualitative comparison of our second-order interaction attributions against first-order variants. A detailed comparison between first-order methods has been presented by Zhao et al. (2024). We closely follow their evaluation protocol and extend it to second-order methods. Unless stated otherwise, we attribute to the second-last hidden representation in the models' image and text encoders and use $N = 50$ integration steps (cf. Equation 9), with a black image as the image reference and a padding token sequence as the text reference. In Appendix D, we include additional experiments on the accuracy of our attributions as a function of $N$, as well as different reference choices.

**Baselines.** We compare our method against four baselines: Interaction-CAM (ICam) (Sammani et al., 2023) is also gradient-based and can be seen as a special case of our approach as shown in Appendix G. Interaction-Lime (ILime) is a bilinear extension of Lime for dual-encoder models (Joukovsky et al., 2023). Code is not available, therefore, we reimplement it; details are in Appendix H. Itsm (Sammani et al., 2023) follows the simple approach of pair-wise multiplication of token and image patch embeddings after applying Clip's final projection layer to the individual embeddings. Originally, it is applied to output representations and we refer to this variant as Itsm-O. We also apply it to the same hidden representations that our method attributes to and refer to this variant as Itsm-H. A qualitative comparison between all methods is included in Figure 11. None of the used methods including our own modifies the model architecture, its parameters, embeddings or gradients.

**Input perturbation.** Following Sammani et al. (2023), we perform conditional perturbation experiments by iteratively removing or inserting the most attributed features in one input while keeping the other input unmodified. Figure 3 plots the decrease in similarity score for Cid. Our method produces the steepest score decline as a function of the number of patches removed, indicating its ability to identify the most relevant interactions. Next to Cid, we also evaluate conditional image patch insertion (Cii) as well as text token deletion (Ctd) and insertion (Cti). All plots are shown in Figures 16 and 17. Table 1 provides a summary and reports the area under the curve (Auc) for the four variants. With the exception of ILime on the text side, our method consistently results in the highest Auc values for the insertion experiments and the lowest for deletion. While ILime performs well on conditional text attribution, interestingly, its image attributions are not competitive. We discuss this in Appendix H. Insertion and deletion experiments have been criticized for producing out-of-domain inputs (Hooker et al., 2019). Therefore, we also construct in-domain perturbations through *hard negative captions* and evaluate their effects in Section 4.3.

---

[1]https://www.kaggle.com/datasets/shtvkumar/karpathy-splits
[2]CLIP family: https://github.com/openai/CLIP, Open family: https://github.com/mlfoundations/open_clip

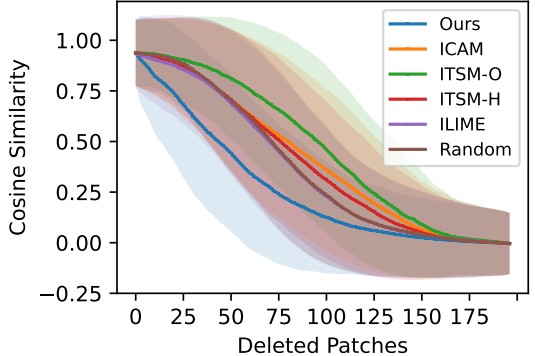

| | Ours | ICAM | ITSM-O | ITSM-H | ILIME | Random |
|---|---|---|---|---|---|---|
| | | | LAION (tuned) | | | |
| CID ↓ | **65.94** | 91.39 | 101.35 | 88.88 | 83.43 | 85.04 |
| CII ↑ | **113.35** | 80.21 | 69.01 | 87.53 | 86.48 | 85.18 |
| CTD ↓ | 4.32 | 7.86 | 6.28 | 7.41 | **4.05** | 7.52 |
| CTI ↑ | 9.26 | 7.77 | 7.82 | 6.72 | **9.89** | 7.47 |
| | | | OPENAI | | | |
| CID ↓ | **16.33** | 21.04 | 23.97 | 20.91 | 20.25 | 20.47 |
| CII ↑ | **24.43** | 20.47 | 16.24 | 20.60 | 20.58 | 20.47 |
| CTD ↓ | **1.06** | 1.35 | 1.15 | 1.30 | 1.07 | 1.31 |
| CTI ↑ | **1.06** | 0.95 | 0.99 | 0.92 | 1.04 | 0.96 |

Figure 3: **Decline of average similarity scores for iterative image patch deletions according to attributions** for the LAION model fine-tuned on COCO. Uncertainty intervals are standard deviation over the evaluation split.

Table 1: **The AUC for CID, CII, CTD, and CTI**, on COCO for the fine-tuned LAION and the original OPENAI model. ↓: lower is better; ↑: higher is better. Corresponding plots in Fig. 16.

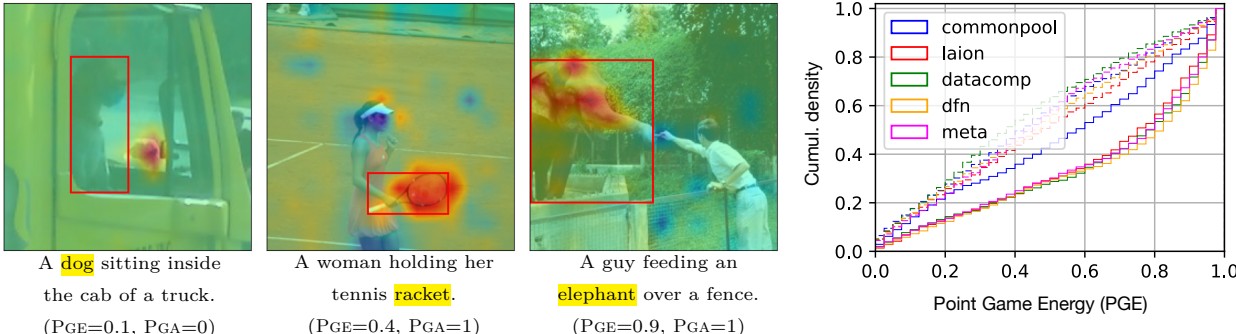

Figure 4: (Left) **Examples for attributions between selected objects in the caption (yellow) and the image** together with corresponding COCO bounding-boxes (red), PGE and PGA values as described in Section 4.2. (Right) **Cumulative PGE distributions** for the OPENCLIP models on COCO before (dashed) and after (solid) in-domain fine-tuning.

**Object localization.** Following Zhao et al. (2024), we employ the Point Game (PG) framework by Zhang et al. (2018) to evaluate how well attributions between objects mentioned within captions and images, correspond to human bounding-box annotations. In FLICKR30K, spans in captions that correspond to bounding-boxes are already annotated. In HNC, object classes exactly match sub-strings in captions and for COCO, we identify objects in captions through a dictionary based synonym matching. For this experiment, we include all object annotations that correspond to a single instance of its class in the image, and whose bounding-box is larger than one patch. This results in 3.5k image-caption pairs from COCO, 8k pairs from FLICKR30K, and 500 pairs from HNC. Within the PG-framework, Point Game Accuracy (PGA) defines the fraction of cases for which the most attributed patch falls within the objects' bounding-box, and Point Game Energy (PGE) is the fraction of positive attributions within the bounding-box relative to the total attribution (Zhao & Chan, 2023; Wang et al., 2020). For PGE, we compare both full distributions (Figure 4 (right)) and median values (mPGE). Figure 4 shows examples from different PGE-ranges and the corresponding cumulative distributions. Very high or low values, unambiguously indicate good correspondence or clear failure cases, respectively. Intermediate values often result from attributions extending to contextual elements beyond actual bounding boxes, such as the *tennis court* in the second example.

Table 2a compares our method against the baselines. Full results are in Table 5. **Our attributions outperform the baselines by large margins.** Figure 15 includes corresponding cumulative PGE-distributions.

|  |  | Coco | | Flickr30k | |
|---|---|---|---|---|---|
| **Training** | **Method** | mPge | Pga | mPge | Pga |
| | Itsm-O | 18.1 | 21.4 | 19.5 | 23.3 |
| | Itsm-H | 29.8 | 38.1 | 29.5 | 37.4 |
| OpenAI | Ilime | 27.9 | 34.9 | 25.8 | 33.1 |
| | Icam | 38.6 | 54.6 | 33.5 | 51.4 |
| | Ours | **72.3** | **79.0** | **64.4** | **72.1** |
| | Itsm-O | 22.8 | 30.3 | 24.5 | 28.7 |
| | Itsm-H | 30.5 | 34.6 | 28.8 | 36.6 |
| Laion (tuned) | Ilime | 28.8 | 37.8 | 25.8 | 34.5 |
| | Icam | 32.5 | 58.4 | 33.5 | 51.4 |
| | Ours | **71.1** | **83.2** | **54.3** | **61.8** |

(a) **PG-based comparison of our attributions against all baselines** described above.

|  |  | Coco | | Hnc | | Flickr30k | |
|---|---|---|---|---|---|---|---|
| **Training** | **Tuning** | mPge | Pga | mPge | Pga | mPge | Pga |
| OpenAI | No | 72.3 | 79.0 | 57.0 | 65.0 | 64.4 | 72.1 |
| | Yes | **78.0** | **82.9** | - | - | **73.4** | **79.0** |
| Laion | No | 49.4 | 63.3 | 40.0 | 51.6 | 38.2 | 52.0 |
| | Yes | **71.1** | **83.2** | - | - | **54.6** | **61.8** |

(b) **Results of the PG-based grounding evaluation** for the OpenAI and Laion models. *Tuning* indicates whether a model was fine-tuned on the respective train split of a dataset. Improvements upon fine-tuning are in bold.

Table 2: Pga: Point Game Accuracy, mPge: median Point Game Energy. Extensive results for Table 2a including additional models are shown in Table 5. Full results of Table 2b can be found in Tables 3 and 4.

Based on these distributions, we test whether improvements are statistically significant using the framework of stochastic order (Dror et al. (2019); details in Appendix E). At the strict criterion of $p < 0.001$ and $\epsilon = 0.01$, our method consistently results in significantly better Pge-statistics.

## 4.3 Model analysis

We now turn to applying our method to gain insights into how Clip models match images and captions.

**Intrinsic grounding ability.** Many of the tested models achieve good performances on the object localization task. On Coco, the off-the-shelf OpenAI (fine-tuned Laion) points to the correct objects in images in 79.0% (83.2%) of the cases (Pga) and their high Pge values show that overall, attributions are distributed to the correct image regions. Table 2b as well as Table 3 and 4 include the results for all models and datasets. We emphasize that all models including the fine-tuned ones, have only been trained on contrastive caption-image matching. Therefore, the strong intrinsic grounding abilities that we observe here show that **the coarse contrastive objective can induce fine-grained correspondence between caption parts and image regions in CLIP models**. However, we also observe large differences between the original models and the fine-tuned variants, especially in the OpenClip models.

**Out-of-domain effects.** The off-the-shelf models were trained on large web-based captioning datasets but have (presumably) not been exposed to the Flickr30k and Coco train splits. To assess domain effects of their grounding abilities, we compare the versions fine-tuned on respective train splits to the original models in Table 2b and the Appendix. Figure 4 also plots cumulative Pge distributions for both. While the unmodified OpenAI model already demonstrates strong grounding abilities on Coco and Flickr30k, the off-the-shelf OpenClip counterparts perform notably worse. Upon in-domain fine-tuning the OpenClip models improve by an average of 21.7±8.3 (14.4±4.3) percentage points (p.p.) in median Pge and by 18.7±6.2 (9.18±4.4) p.p. in Pga on Coco (Flickr30k). All changes are significant at $p < 0.001$ and $\epsilon = 0.01$. These large improvements indicate that **this fine-grained connection between captions and images, however, struggles to generalize beyond the training domain**.

**Class-wise evaluation.** The generalization issue of the models' grounding ability becomes apparent in the examples shown in Figure 5. The off-the-shelf model fails to identify the *clock* and even assigns a negative attribution to the *surfboard*, whereas the fine-tuned version clearly identifies both. To examine the models' understanding of individual visual-linguistic concepts in more detail, we break the above analysis down to individual classes. The right side of Figure 5 shows average Pge-values and their standard deviations for Coco classes in the OpenClip Laion model. The classes are ordered from left to right based on their average grounding ability in the unmodified model (blue). The model effectively identifies the leftmost classes,

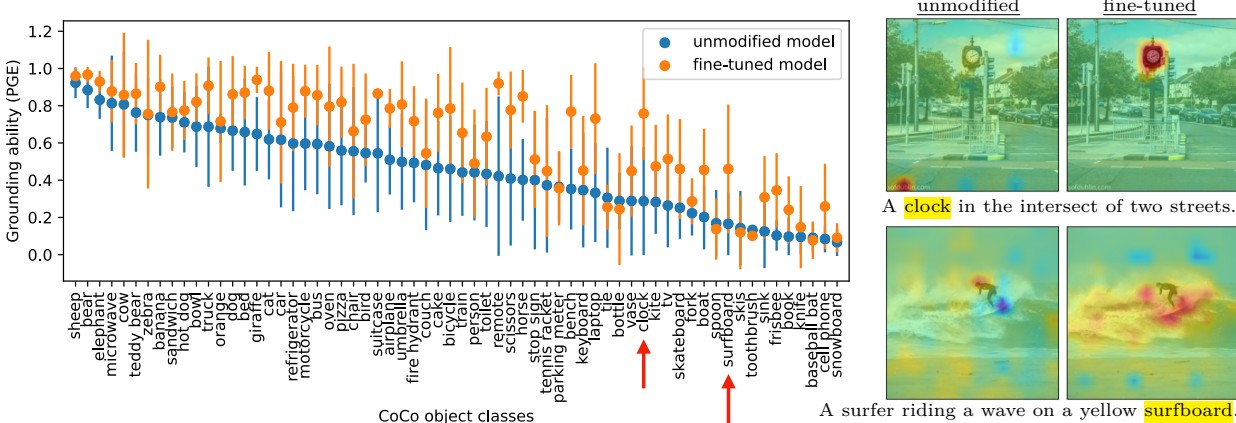

Figure 5: (Left) **Class-wise average PGE before and after in-domain fine-tuning** in the OPEN-CLIP LAION model on COCO. Error bars are standard deviations over all class instances. (Right) Two explicit examples of how the model's grounding ability changes upon tuning. The corresponding classes are emphasized with red arrows on the left.

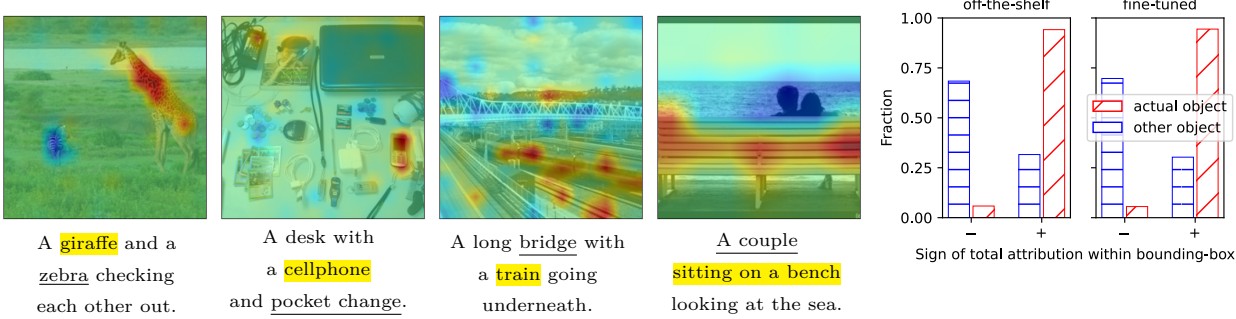

Figure 6: (Left) **Examples of negative attributions for mismatches**. Attributions are for yellow selections in captions. Mismatching objects (underlined) receive negative attributions (blue). The histogram on the right shows the distribution over the sign of such cross-attributions.

while grounding is notably weaker for the rightmost. Upon fine-tuning (orange), most classes show clear improvements. By means of the standardized mean difference between the two PGE values, we observe the largest improvements for the classes *horse*, *bench*, *giraffe*, *airplane* and *clock*. This shows that **contrastive fine-tuning can sharpen the visual-linguistic conception of individual object classes**. In Appendix C (Figure 18), we replicate this experiment for DFN and COMMONPOOL yielding similar results.

**Object Discrimination.** We frequently observe that attributions between a given object in the text and a non-matching one in the image – or vice versa – are not only neutral but negative. Figure 6 includes four explicit examples. To systematically evaluate this effect, we sample instances from COCO that include at least two distinct object classes, each appearing exactly once in the image. We then compute attributions between the two corresponding bounding-boxes and text spans and also across them, which we refer to as cross-attribution. Attribution to the actual object's bounding-box is positive in 94.1% (94.1%) of all cases, while cross-attributions to the other object are negative in 68.4% (69.7%) of instances in the original (fine-tuned) model (cf. Figure 6 (right)). This implies that **CLIP models do not only match corresponding objects across the input modes but can actively penalize mismatches by assigning them negative contributions**.

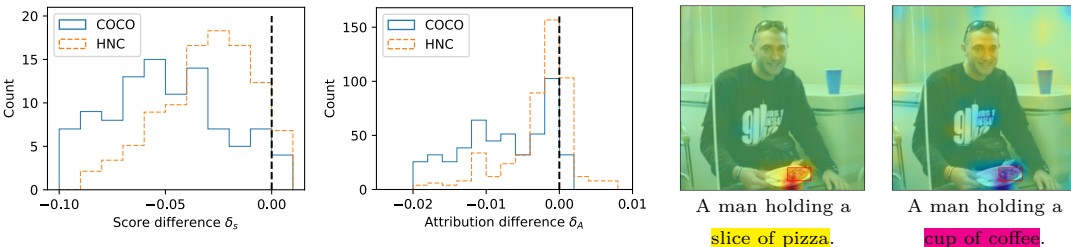

Figure 7: **Attribution changes in hard negative captions**. (Left) Histograms for score ($\delta_S$) and attribution ($\delta_A$) differences. (Right) An example with the true caption on the left and a hard negative caption on the right. The true object is marked in yellow, and the replaced negative one in magenta.

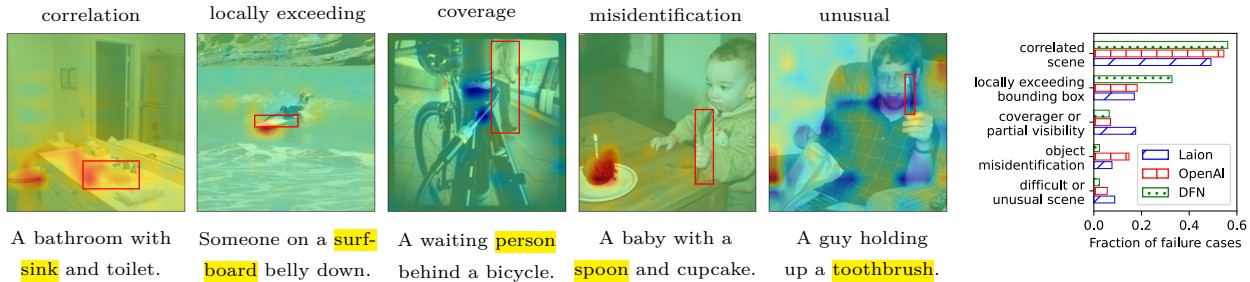

Figure 8: **Examples for the five failure categories** that we can identify (left) and their relative occurrence in three models (right). More examples for all categories are in included in Figure 12.

**Hard negative captions.** On the text side, it is straightforward to produce in-domain perturbations. We create hard negative captions that replace a single object in a positive caption with a reasonable but different object to receive a negative counterpart. To this end, we leverage the automatic procedure by Dönmez et al. (2023) together with our simplified template (cf. Section 4) and additionally create a second resource from Coco by manually annotating a small yet high-quality evaluation sample of 100 image-caption pairs.

We check whether our negative captions actually result in a decrease of the predicted similarity score compared with their positive counterparts and define the difference as $\delta_S$. It is negative in 95.2% (89.1%) of the Coco (Hnc) pairs. We then compute attributions between the token range of the original or replaced object and the object bounding-box in the image and define the attribution difference as $\delta_A$. It is also negative in 95.2% (74.1%) of the Coco (Hnc) examples. Full histograms for $\delta_S$ and $\delta_A$ as well as an example for a change in attributions is included in Figure 7. These results show that **the model mostly reacts correctly to mistakes in captions and decreases the correspondence between affected image regions and caption spans**.

**Qualitative failure analysis.** To identify cases where the models' grounding abilities are systematically weak, we extract objects with Pge $< 0.2$ from the Coco validation set and categorize them qualitatively. For the Laion , OpenAI , and Dfn models, this results in approximately 200 image-caption pairs each. **We can identify five major failure categories**: (1) Visually correlated scenes like baseball courts, bathrooms, offices, etc., (2) attributions locally exceeding bounding boxes, (3) coverage or partial visibility of objects, (4) actual object misidentifications, and (5) difficult or unusual scenes. Figure 8 shows the distribution among these categories and an example for each. More examples are included in Figure 12. Category (1), correlated scenes, accounts for approximately half of all failures in all three models, indicating that Clip models may struggle to differentiate between objects that commonly appear together.

## 5 Discussion

**Interpretation of results and future work.** While prior work has already established that CLIP models can ground full text inputs onto images and vice versa (Zhao et al., 2024), our second-order attributions take these insights a step further and show that this visual-linguistic correspondence is more fine-grained connecting individual parts of captions and images.

Our evidence for this intrinsic grounding ability to be significantly reduced on data outside the initial training domain complements recent efforts towards an understanding of CLIP's ostensible out-of-domain generalization (Xue et al., 2024). While Fang et al. (2022) and Nguyen et al. (2022) identified the training distribution as the critical component, Mayilvahanan et al. (2024; 2025) recently showed that it must be assigned to domain contamination of web-scale training datasets and CLIP models do *not* actually generalize to unseen image domains, like renditions.

The finding that CLIP models can actively assign negative contributions to mismatches reveals a non-trivial mechanism in their prediction computation. The fact that this is not consistently the case and we also observe positive cross-attributions in correlated scenes like tennis courts, bathrooms, kitchens, streets, etc., however, is yet to be understood. It suggests the contrastive objective may not provide sufficient supervision to learn to tell apart objects that frequently co-occur. Future work should establish a detailed understanding of this phenomenon. A solution may be to augment the training data with negatives targeting such correlations (Yuksekgonul et al., 2023; Patel et al., 2024).

Our baseline experiments show that analyzing interactions in CLIP models is not trivial. Neither simplified gradient-based approaches (ICAM), pair-wise embedding multiplication (ITSM) nor surrogate modeling (ILIME) are sufficient for the purpose. It may still be possible to further enhance our method accounting for discrete text representations (Sanyal & Ren, 2021), incorporating non-uniform interpolation (Bhat & Raychowdhury, 2023) or integrating along non-linear paths (Kapishnikov et al., 2021; Zhuo & Ge, 2024).

**Limitations.** As stated explicitly in Equation 11, our interaction attributions are an approximation. Throughout this work, we attribute to intermediate representations of inputs, which is both efficient and informative (Möller et al., 2023). In transformers, intermediate representations have undergone multiple contextualization steps and are technically not strictly tied to input features at a given position. Finally, recently proven fundamental limitations of attribution methods urge caution in their interpretation especially regarding counterfactual conclusions about feature importance (Bilodeau et al., 2024).

Despite these considerations, our consistent results on caption-image interactions across a variety of models and datasets provide strong empirical evidence for the evolvement of fine-grained inter-modal correspondence in CLIP models through contrastive training. While we must be more careful drawing definite conclusions about specific failure cases, we argue that explainability methods like ours can be used to formulate hypotheses about mistakes and biases in models. We cannot regard them as guaranteed robust and faithful, but they provide insights that have the potential to improve models further.

## 6 Conclusion

We derive general second-order attributions in dual encoder architectures, enabling the attribution of similarity predictions onto interactions between input features. Our method is applicable to any differentiable dual-encoder and requires no modifications of the initial model. We believe it can also provide valuable insights into more complex relations between images and text (Krishna et al., 2017), models for different modalities (Guzhov et al., 2022) and applications like retrieval (Mueller & Macdonald, 2025; Vast et al., 2024). Our experiments with CLIP models provide strong evidence for them capturing fine-grained interactions between corresponding visual and linguistic concepts despite their coarse contrastive objective. At the same time, we also observe pronounced out-of-domain effects. These results complement recent findings identifying limitations in the generalization capabilities of CLIP (Mayilvahanan et al., 2025). Finally, an error analysis revealed that CLIP models can struggle with covered or partially visible objects, unusual scenes, and correlated contexts like kitchens, offices, or sports courts.

By enabling the analysis of interactions between caption and image features, our approach contributes to an emerging interest in understanding higher-order dependencies in CLIP models (Gandelsman et al., 2025; Joukovsky et al., 2023), reaching beyond well-understood first-order effects (Zhao et al., 2024).

## Acknowledgements

Funded by Deutsche Forschungsgemeinschaft (DFG, German Research Foundation) under Germany's Excellence Strategy - EXC 2075 – 390740016. We acknowledge the support by the Stuttgart Center for Simulation Science (SimTech).

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

## A Additional Examples

Figure 9 shows two additional examples for inter-modal attributions, one for text-span selection and image projection and one for bounding-box selection and caption projection.

Figure 11 shows a qualitative comparison between our attributions and the baselines described in Section 4.2.

Figure 12 shows five examples for each of the five failure categories that we identified in Section 4.3 under *Qualitative failure analysis*.

## B Intra-modal attributions

. This section describes intra-modal model attributions for text- or image-pairs exemplified in Figure 2 For text-text attributions, after summation over embedding dimensions, the attributions take the form of an $S_1 \times S_2$ dimensional matrix, with $S_1$ and $S_2$ being token sequence lengths of the two texts. For image-image pairs, attribution tensors become four dimensional taking the shape $(H \times W)_1 \times (H \times W)_2$, containing a contribution for every pair of two patches from either image. Figure 10 includes additional examples.

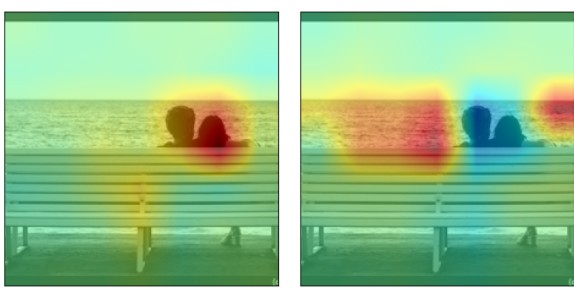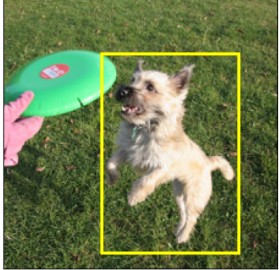

(a) A couple sitting on a bench looking at the sea. (b) A couple sitting on a bench looking at the sea. (c) A dog is jumping for a frisbee. (d) A dog is jumping for a frisbee.

Figure 9: Additional examples for inter-modal attributions of token-range selection with image projections (left) and bounding-box selection with caption projection (right). The visualization is identical to Figure 1.

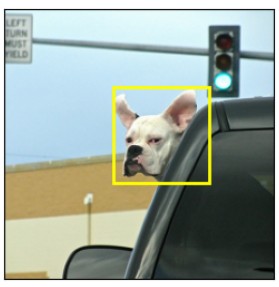 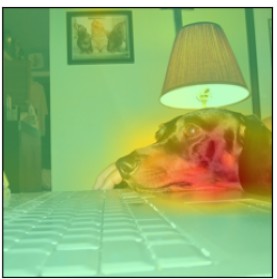 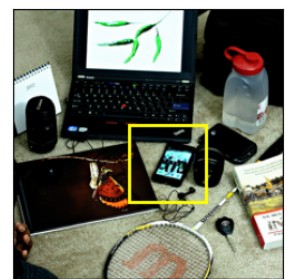 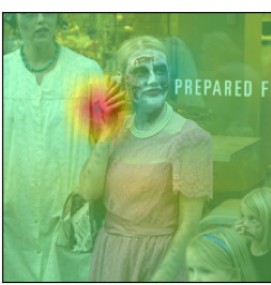

Figure 10: Image-image attributions between the yellow bounding-box in the left image and the one to its right as described in Section 3. Visualisation is identical to Figure 2 (right)

| Model | Tuning | Coco | | | HNC | | | Flickr30k | | |
|---|---|---|---|---|---|---|---|---|---|---|
| | | mPGE | PGE>0.8 | PGA | mPGE | PGE>0.8 | PGA | mPGE | PGE>0.8 | PGA |
| RN50 | No | 66.3 | 28.8 | 76.9 | 50.1 | 22.6 | 61.8 | 60.1 | 25.5 | 71.2 |
| ViT-B/32 | No | 63.5 | 33.3 | 69.1 | 52.8 | 28.5 | 58.5 | 50.4 | 23.4 | 58.1 |
| ViT-B/16 | No | 72.3 | 35.7 | 79.0 | 57.0 | 31.7 | 65.0 | 64.4 | 28.4 | 72.1 |
| ViT-B-16 | Yes | **78.0** | **48.4** | **82.9** | - | - | - | **73.4** | **40.7** | **79.0** |

Table 3: Summary of the Point-Game evaluation for different CLIP models by OpenAI as described in Section 4.2. *Model* refers to the investigated architecture, *Tuning* is whether the model was fine-tuned on the train split of the respective dataset. Best overall results are in bold, best results of unmodified models are underlined.

## C   Extended results

Table 3 shows full results for our Point-Game evaluation on different OpenAI models. Next to the ViT-B/16 architecture, we also evaluate the RN50 and ViT-B/32 variants. Table 4 includes the full evaluation for all OpenClip models. In addtion to the median Pge (mPGE), in these tables we also report cumulative Pge densitites for the $80^{th}$ percentile (Pge>0.8). Full cumulative Pge-histograms for additional models are included in Figures 13 and 14.

Table 5 presents full results of our Point-Game baseline experiments extending Section 4.2. Corresponding cumulative densities of the PGE-metric are shown in Figure 15. Figures 16 and 17 show the plots of the conditional insertion and deletion experiments for the OpenClip Laion model and the original OpenAI model, respectively. The corresponding Auc values are contained in Table 1.

Figure 18 extends the class-wise PGE-evaluation from Section 4.3 to the OpenClip Dfn and Data-Comp models.

## D   Additional Experiments

**Approximation Error.**   In Section 3, we have shown the equality between Eq. 2 and Eq. 10. The only approximation affecting this equality is the numerical integration in Eq. 9 to calculate the two *integrated Jacobians* $\mathbf{J}^g$ and $\mathbf{J}^h$ by a sum over $N$ bins. We can evaluate how good this approximation is by explicitly calculating the four similarity predictions between the references and inputs $f(\mathbf{a},\mathbf{b})$, $f(\mathbf{r}_a,\mathbf{b})$, $f(\mathbf{a},\mathbf{r}_b)$ and $f(\mathbf{r}_a,\mathbf{r}_b)$, as well as the attribution matrix $\mathbf{A}$. The *approximation error* can then be defined as the absolute difference between Eq. 2 and Eq. 10. In Figure 19, we plot this error as a function of different magnitudes for $N$. For larger $N$, it converges as expected.

**Choice of references.**   The choice of the references $\mathbf{r}_a$ and $\mathbf{r}_b$ is ambiguous as long as they are uninformative. We try different options and evaluate their approximation errors as defined above. For the image input,

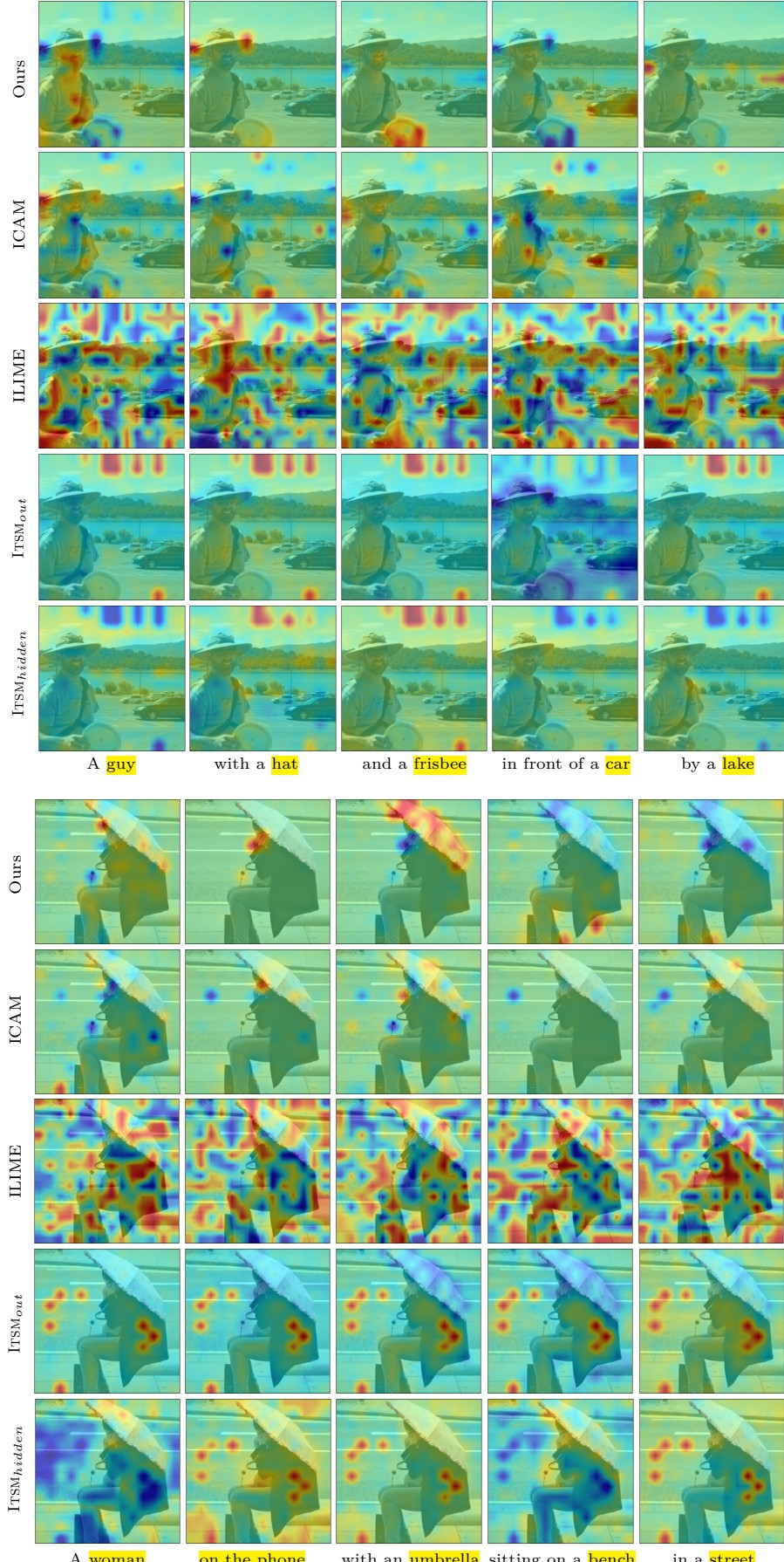

Figure 11: Qualitative comparison between our attributions, the ICAM, ILIME and both ITSM variants. Heatmaps over images in a given column are for the marked parts of the captions in yellow below.

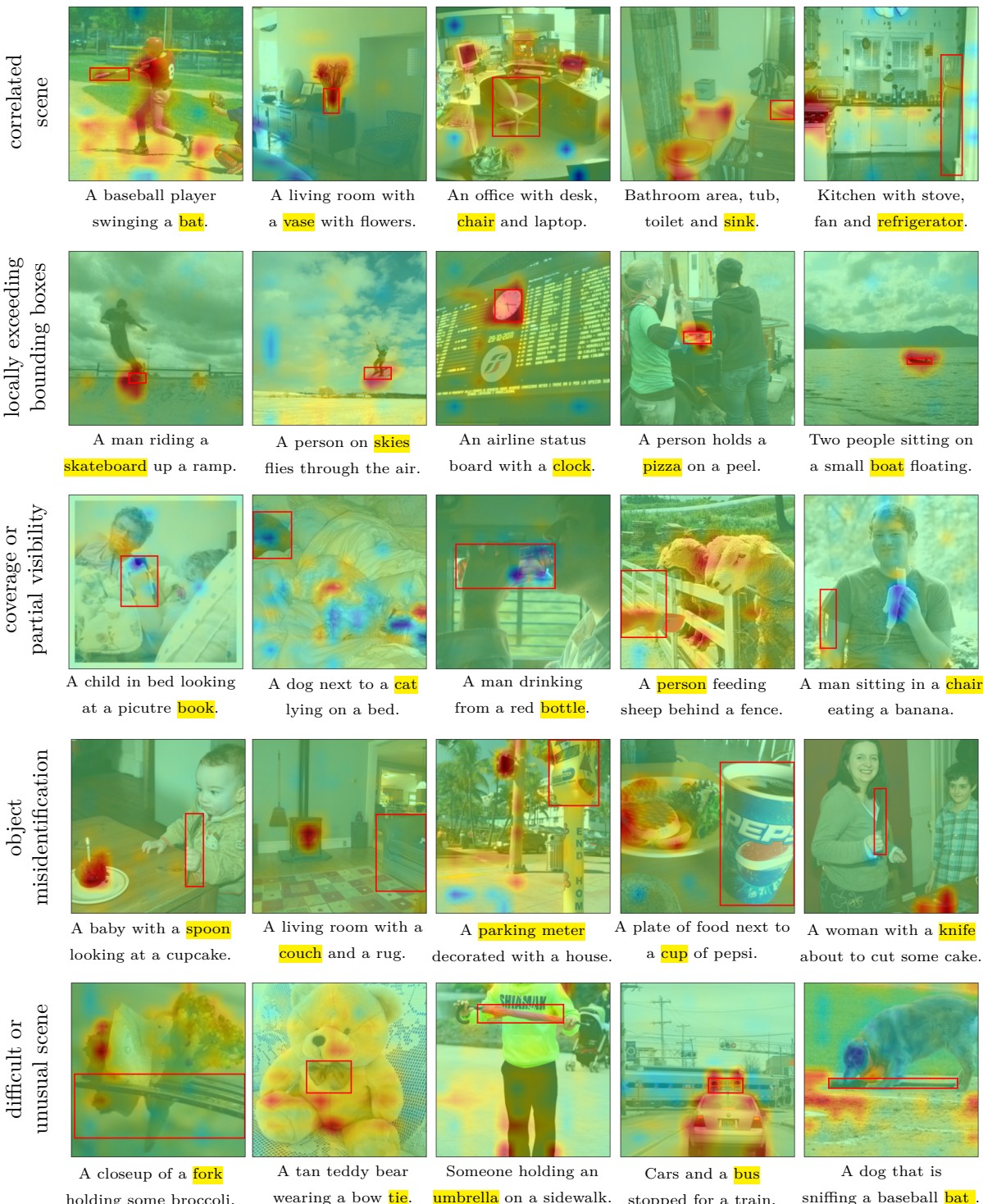

Figure 12: Five examples for each of the five identified failure categories as described in Section 4.3.

|  |  | COCO | | | FLICKR30K | | |
|---|---|---|---|---|---|---|---|
| **Training** | **Tuning** | mPGE | PGE>0.8 | PGA | mPGE | PGE>0.8 | PGA |
| LAION | No | 49.4 | 22.0 | 63.3 | 38.2 | 15.9 | 52.0 |
| | Yes | 71.1 | 47.3 | **83.2** | **54.6** | 30.6 | **61.8** |
| COMMONPOOL | No | 43.0 | 18.2 | 58.8 | 36.7 | 15.5 | 53.0 |
| | Yes | 57.7 | 28.7 | 67.1 | 44.6 | 20.8 | 56.2 |
| DATACOMP | No | 38.5 | 14.6 | 56.0 | 32.8 | 11.8 | 48.9 |
| | Yes | **72.4** | 50.0 | 75.1 | 50.7 | 27.3 | 56.0 |
| DFN | No | 46.5 | 19.6 | 54.3 | 35.4 | 12.3 | 43.3 |
| | Yes | 71.4 | **53.3** | 74.6 | 53.1 | **33.5** | 58.3 |
| Meta-CLIP | No | 44.2 | 16.8 | 52.3 | 37.0 | 14.5 | 46.4 |
| | Yes | 57.5 | 49.8 | 77.1 | 49.2 | 24.1 | 57.2 |

Table 4: Summary of the Point-Game evaluation for all OPENCLIP models on COCO and Flickr30k. The *Training* column refers to the dataset the model was initially trained on, *Tuning* is whether the model was additionally fine-tuned on the train-split of the respective evaluation dataset. All models implement the ViT-B-16 architecture except Meta-CLIP that uses quickgelu activations. Best overall results are in bold, best results for unmodified models are underlined.

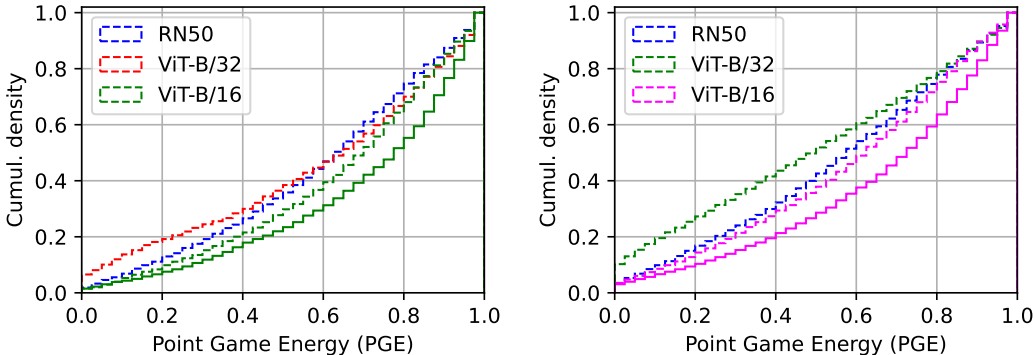

Figure 13: Cumulative PGE-distribution plots of the unmodified (dashed) / fined-tuned (solid) OPE-NAI models on COCO (left) and FLICKR30K (right) dataset as described in Section 4.2.

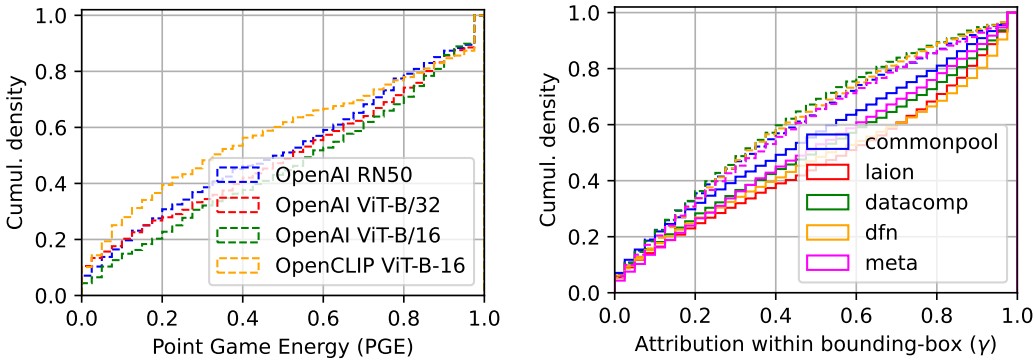

Figure 14: Cumulative PGE-distribution plots for the different models on HNC (left) and the unmodified (dashed) / fine-tuned (solid) OPENCLIP models on FLICKR30K (right) as described in Section 4.2.

we use a black image and zero-centered gaussian noise. For the text input, we use a sequence of padding tokens and the empty sequence consisting only of the `CLS` and `EOS` tokens. Figure 19 includes approximation

| Training | Method | Coco | | Flickr30k | |
|---|---|---|---|---|---|
| | | mPGE | PGA | mPGE | PGA |
| OpenAI | ITSM$_{out}$ | 18.1 | 21.4 | 19.5 | 23.3 |
| | ITSM$_{hidden}$ | 29.8 | 38.1 | 29.5 | 37.4 |
| | ILIME | 27.9 | 34.9 | 25.8 | 33.1 |
| | ICAM | 38.6 | 54.6 | 33.5 | 51.4 |
| | Ours | **72.3** | **79.0** | **64.4** | **72.1** |
| LAION (tuned) | ITSM$_{out}$ | 22.8 | 30.3 | 24.5 | 28.7 |
| | ITSM$_{hidden}$ | 30.5 | 34.6 | 28.8 | 36.6 |
| | ILIME | 28.8 | 37.8 | 25.8 | 34.5 |
| | ICAM | 32.5 | 58.4 | 33.5 | 51.4 |
| | Ours | **71.2** | **83.2** | **56.3** | **63.6** |
| DFN (tuned) | ITSM$_{out}$ | 24.2 | 34.5 | 25.1 | 31.4 |
| | ITSM$_{hidden}$ | 27.4 | 23.0 | 27.7 | 36.5 |
| | ILIME | 27.9 | 39.2 | 25.7 | 33.5 |
| | ICAM | 33.3 | 46.5 | 24.2 | 42.2 |
| | Ours | **71.4** | **74.6** | **53.1** | **58.3** |
| DATACOMP (tuned) | ITSM$_{out}$ | 25.5 | 38.7 | 26.5 | 33.9 |
| | ITSM$_{hidden}$ | 35.0 | 42.3 | 22.6 | 28.4 |
| | ILIME | 28.4 | 39.3 | 25.7 | 34.1 |
| | ICAM | 36.9 | 49.5 | 23.2 | 37.3 |
| | Ours | **72.4** | **75.1** | **50.7** | **60.0** |

Table 5: PGE-evaluation results of our method compared against the ITSM and InteractionCAM (ICAM) baselines for different models as described in Section 4.2 under *Object localization.* Best results for every model are in bold.

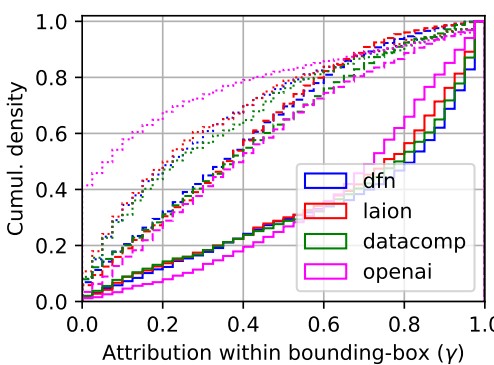

Figure 15: Cumulative PGE-distributions for our baseline experiment in Section 4.2. Our method is in solid, InteractionCAM is dashed and ITSM$_{out}$ is dotted. ITSM$_{hidden}$ is excluded for an uncluttered visualization.

errors for all four combinations of these references as a function of $N$. Combinations with gaussian noise for the image reference appear to converge slightly faster than the black image.

For different references, attributions can vary slightly. However, these differences are small, even for large objects like the cow in Figure 20 (left), for which attributions tend to spread out. The absolute difference of attributions between any two combinations of references is on the order of $10^{-5}$ and tends to decrease for larger $N$. The plot in Figure 20 (right) shows this for the difference between a *black-padding* and *gaussian-empty* reference combination.

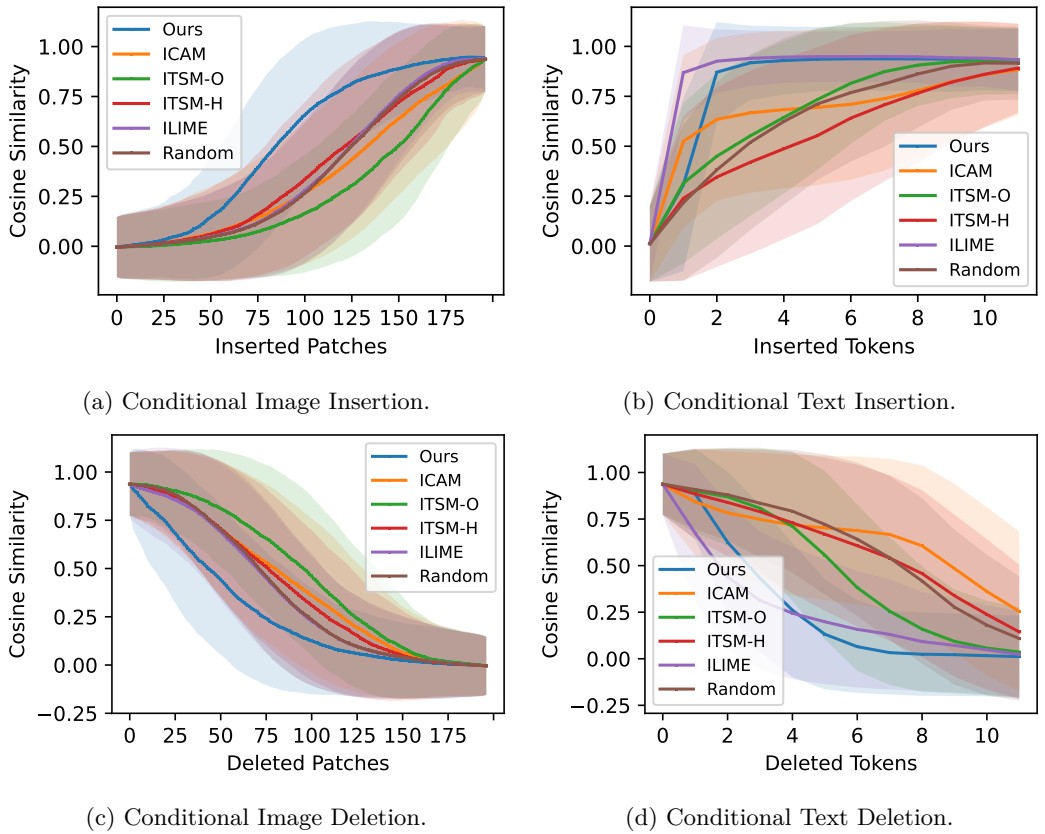

(a) Conditional Image Insertion.

(b) Conditional Text Insertion.

(c) Conditional Image Deletion.

(d) Conditional Text Deletion.

Figure 16: Average change in similarity score upon conditional insertion and deletion performed on either the caption or the image using a ViT-B-16 model pretrained on LAION . Confidence intervals are standard deviations over the evaluation dataset. Table 1 summarizes the AUC of these plots.

## E   Stochastic Dominance

Stochastic dominance defines an order relation between probability distributions based on their cumulatives. del Barrio et al. (2018) have proposed a significance test building on the principle and Dror et al. (2019) have identified it as being particularly suitable to compare deep neural models. The test's $\epsilon$-parameter is the maximal percentile range where the inferior distribution is allowed to dominate the superior one and Dror *et al.* suggest to set it to $\epsilon < 0.4$. The smaller $\epsilon$, the stricter the criterion. $\alpha$ is the significance level.

## F   Integrated Gradients

We derive IG for a model $f(\mathbf{a}) = s$ with a vector-valued input $\mathbf{a}$ and a scalar prediction $s$, e.g. a classification score. We define the reference input $\mathbf{r}$, begin from the difference between the two predictions and reformulate it as an integral over the integration variable $\mathbf{x}$:

$$f(\mathbf{a}) - f(\mathbf{r}) = \int_{\mathbf{r}}^{\mathbf{a}} \frac{\partial f(\mathbf{x})}{\partial \mathbf{x}_i} d\mathbf{x}_i \tag{12}$$

Again we do not write out sums over double indices. To solve the resulting line integral, we substitute with the straight line $\mathbf{x}(\alpha) = \mathbf{r} + \alpha(\mathbf{a} - \mathbf{r})$ and pull its derivative $\partial \mathbf{x}(\alpha)/\partial \alpha = (\mathbf{a} - \mathbf{r})$ out of the integral:

$$\int_0^1 \frac{\partial f(\mathbf{x}(\alpha))}{\partial \mathbf{x}_i(\alpha)} \frac{\partial \mathbf{x}_i(\alpha)}{\partial \alpha} d\alpha = (\mathbf{a} - \mathbf{r})_i \int_0^1 \nabla_i f(\mathbf{x}(\alpha)) \, d\alpha \tag{13}$$

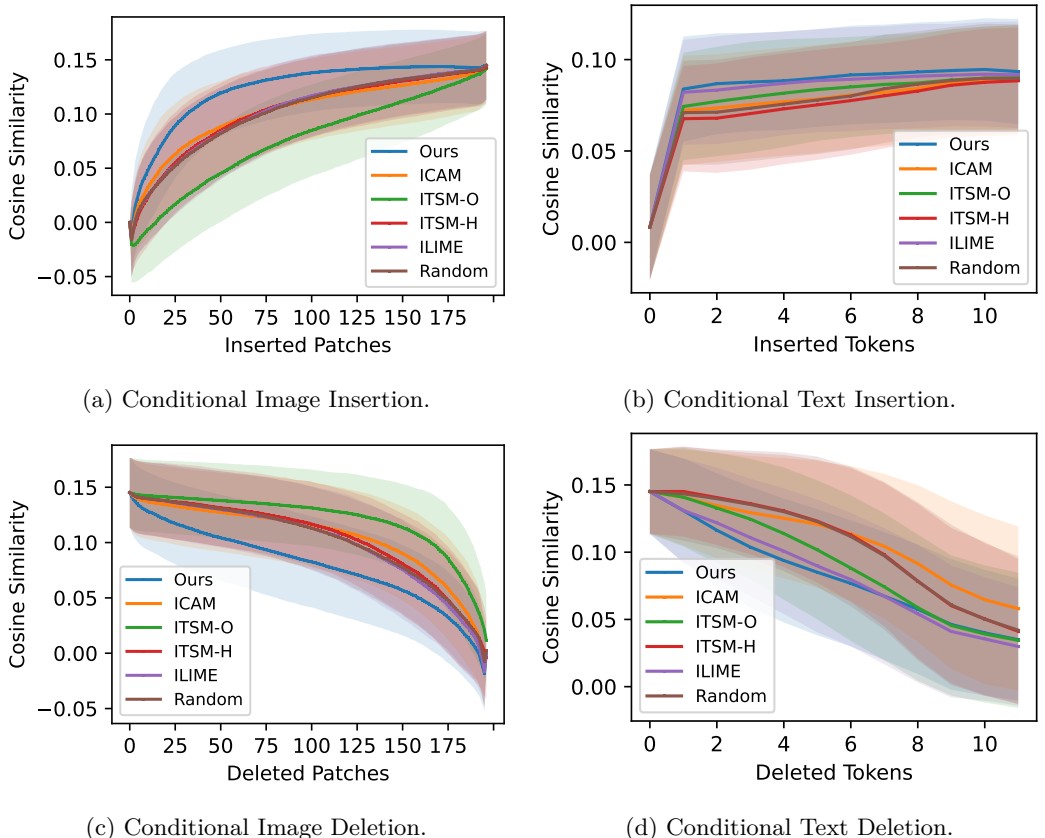

(a) Conditional Image Insertion.

(b) Conditional Text Insertion.

(c) Conditional Image Deletion.

(d) Conditional Text Deletion.

Figure 17: Conditional insertion and deletion performed on either the caption or the image using the original ViT-B/16 model by OPENAI without fine-tuning.

In practice, we approximate the integral by a sum over $N$ steps. If the reference is uninformative, so that $f(\mathbf{r}) \approx 0$, the equality between Eq. 12 and Eq. 13 can be reduced to the final approximation of IG:

$$f(\mathbf{a}) \approx (\mathbf{a} - \mathbf{r})_i \frac{1}{N} \sum_{n=1}^{N} \nabla_i f(\mathbf{x}(\alpha_n)), \tag{14}$$

which decomposes the model prediction $f(\mathbf{a})$ into contributions of individual feature $i$ in $\mathbf{a}$.

## G Relation to interactionCAM

Here, we first discuss the relation of integrated gradients IG and GRADCAM and then show how our second-order method can be reduced to the ICAM baseline.

We start from the right-hand-side of Equation 14, the final form of IG. we can reduce these this result further by setting $N = 1$ and using the zero vector as a reference, (r=0. These simplifications yield,

$$\mathbf{a}_i \nabla_i f(\mathbf{a}), \tag{15}$$

which is often referred to as *gradient×input* and is the basic form of GradCam. The method typically attributes to deep image representations in CNNs, so that $\mathbf{a}$ has the dimensions $C \times H \times W$, the number of channels, height and width of the representation. To reduce attributions to a two-dimensional map, it sums

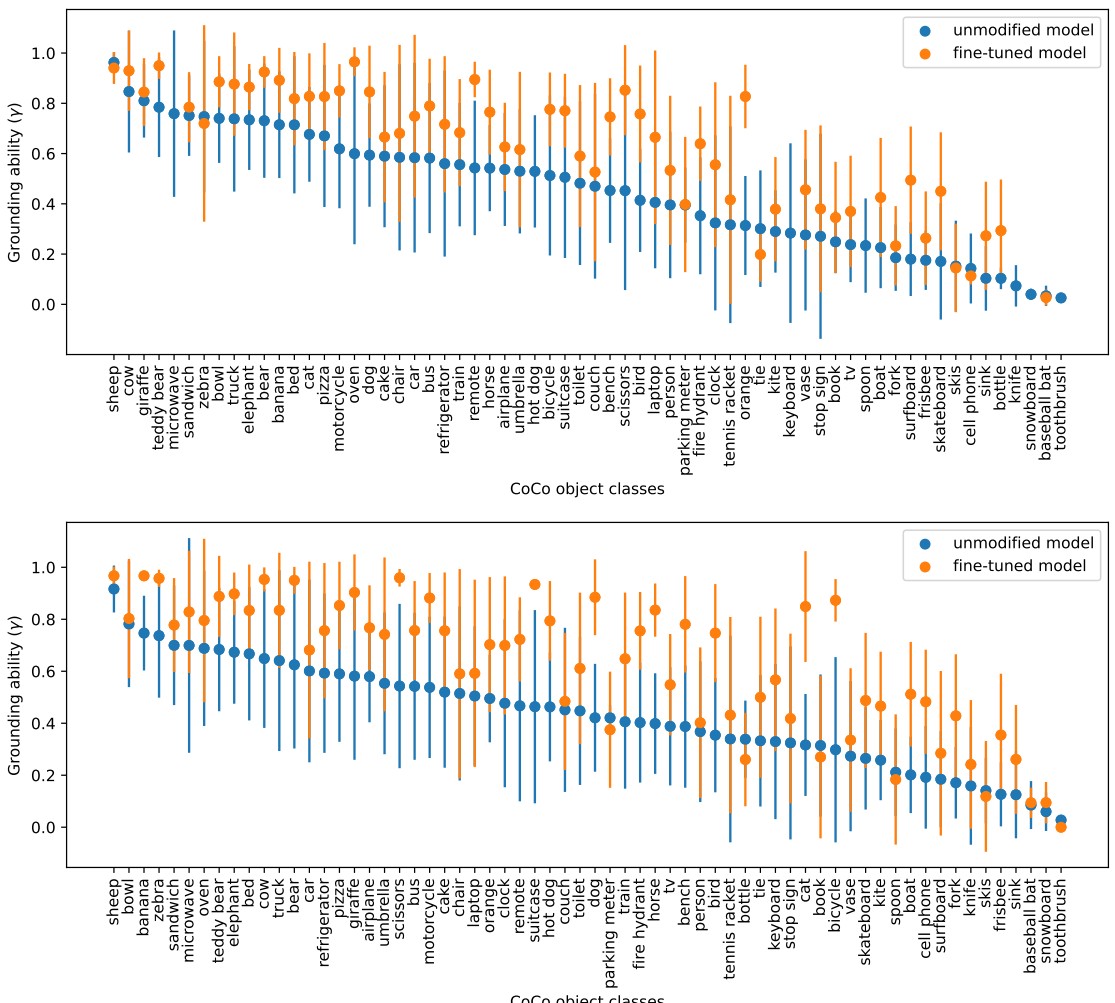

Figure 18: Class-wise PGE-evaluation for the OPENCLIP LAION (top) and DATACOMP (bottom) models before and after in-domain fine-tuning as discussed in Section 4.2.

over the channel dimension and applies a relu-activation to the outcome. The original version also average pools the gradients over the spatial dimensions, however, this is technically not necessary.

As discussed earlier, neither integrated gradients nor GRADCAM can explain interaction in dual encoder predictions. Following the logic from above we can, however, reduce our second-order attributions from Eq. 10 by setting $N = 1$ in the computation of the integrated Jacobians in Eq. 9 and using $\mathbf{r}_a = \mathbf{r}_b = \mathbf{0}$. For our attribution matrix from Equation 10 we then receive the simplified version

$$\mathbf{a}_i \, \frac{\partial \mathbf{g}_k}{\partial \mathbf{a}_i} \, \frac{\partial \mathbf{h}_k}{\partial \mathbf{b}_j} \, \mathbf{b}_j. \tag{16}$$

This simplification could be termed *Jacobians×inputs* and is equivalent to the ICAM by Sammani et al. (2023). Note, however, that setting $N = 1$ is the worst possible approximation to the integrated Jacobians. Therefore, it is not surprising that empirically this version performs worse than our full attributions.

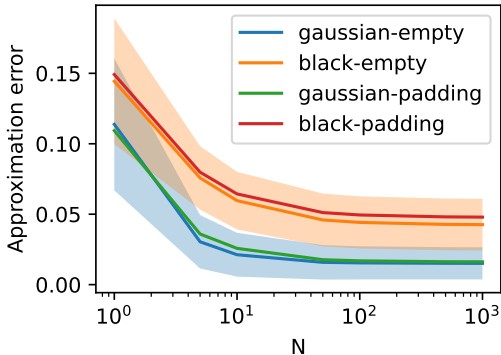

Figure 19: Approximation errors for different reference choices as a function of the number of integration steps $N$. The image references are abbreviated as 'gaussian' for gaussian noise and 'black' for the black image. Text references are 'padding' and 'empty' for a padding sequence and the empty sequence, respectively. Exemplatory standard deviations over the evaluation sample are shown as shades of the respective plots.

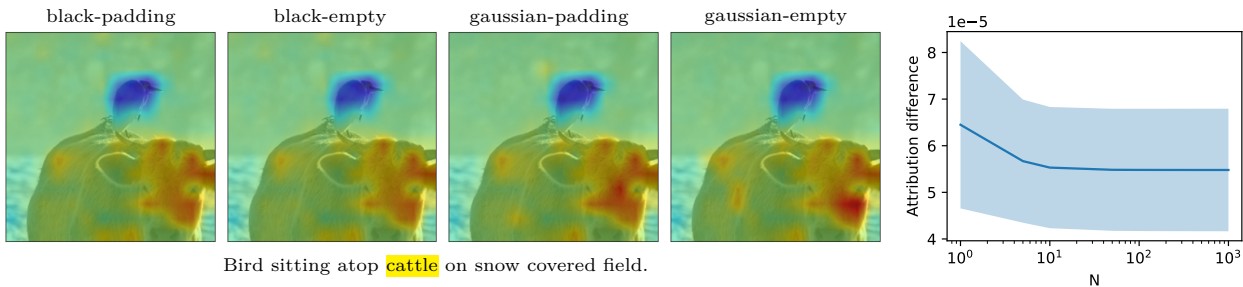

Figure 20: (Left) Attribution differences for all four combinations of references as indicated above the images for the yellow selection in the caption below. (Right) The mean absolute attribution difference between a *black-padding* and a *gaussian-empty* reference combination as a function of the number of approximation steps $N$ and its standard deviation over the evaluation smaple (blue shade).

## H  Interaction LIME

We reimplement the ILIME method proposed by Joukovsky et al. (2023) that extends the principle of LIME Ribeiro et al. (2016) to dual encoder models with two inputs.

The core idea of LIME is to locally approximate the actual model $f$ around a given input with an interpretable surrogate model $\varphi$. The local neighborhood of the input is approximated by a sample of perturbations. The surrogate model is typically linear and operates on latent representations of the input. Further, there needs to be a mapping from latent representations to input representations, so that we can generate corresponding inputs that the actual model can process.

In the image domain latent representations $\mathbf{z}^a$ are typically binary variables indicating the presence or absence of super pixels in the input. To enable a direct comparison to our method and the other baselines, we use the vision transformer's patches as super pixels. Analogously, in the text input we define latent representations $\mathbf{z}^b$ as binary variables indicating the presence of input tokens. Disabled image patches are replaced with the mean over the image, disabled tokes are replaced with the padding token.

The local neighborhood of a given input pair $(\mathbf{a}, \mathbf{b})$ is approximated by sampling $N$ such latent representations $(\mathbf{z}_i^a, \mathbf{z}_i^b)$ from two Bernoulli distributions. For the corresponding input perturbations $(\mathbf{a}_i, \mathbf{b}_i)$, we then compute the CLIP scores $s_i = f(\mathbf{a}_i, \mathbf{b}_i)$ and fit the surrogate model to reproduce these predictions.

To account for interactions between the two inputs in dual encoder models, Joukovsky et al. (2023) propose to use a bilinear form as surrogate model:

$$\varphi(\mathbf{z}^a, \mathbf{z}^b) = \mathbf{z}^{a\top}\mathbf{W}\mathbf{z}^b + c, \tag{17}$$

with a weight matrix $\mathbf{W}$ and a scalar bias $c$, which is then optimized according to the following MSE objective:

$$\min_{\mathbf{W},c} \sum_{i=1}^{N} \pi(\mathbf{a}, \mathbf{a}_i, \mathbf{b}, \mathbf{b}_i) \left( f(\mathbf{a}_i, \mathbf{b}_i) - \varphi(\mathbf{z}_i^a, \mathbf{z}_i^b; \mathbf{W}, c) \right)^2 \tag{18}$$

Here, $\pi$ is a function that weights individual neighborhood samples $(\mathbf{a}_i, \mathbf{b}_i)$ according to their similarity to the original input $(\mathbf{a}, \mathbf{b})$. We use the cosine similarities between perturbed and original captions and image inputs, respectively, and following Joukovsky et al. (2023), define the total similarity weight as the average of the caption and image similarity:

$$\pi(\mathbf{a}, \mathbf{a}_i, \mathbf{b}, \mathbf{b}_i) = \frac{1}{2}\left(\mathbf{g}^\top(\mathbf{a})\,\mathbf{g}(\mathbf{a}_i) + \mathbf{h}^\top(\mathbf{b})\,\mathbf{h}(\mathbf{b}_i)\right) \tag{19}$$

To fit $\varphi$, we use stochastic gradient descent with a learning rate of $10^{-2}$ and weight-decay of $10^{-3}$ over $N = 1000$ samples with Bernoulli drop-out probabilities of $p = 0.3$ for both caption and image representations. These parameters closely align with Joukovsky et al. (2023). Additionally, we find that scaling the latent representations $\mathbf{z}^a$ and $\mathbf{z}^b$ with the square root of the numbers of tokens $\sqrt{S}$ and image patches $\sqrt{H \times W}$, respectively, helps to stabilize convergence.

Finally, the fitted weight matrix $\mathbf{W}$ models interactions between image patches and caption tokens. Therefore, we can evaluate and visualize it in the same way as our attribution matrices $\mathbf{A}$.

In Section 4.2 we found that ILIME performs well – and even slightly better than our method – on conditional caption attribution. At the same time its conditional image attributions are not competitive. Consequently, its grounding ability as evaluated by the PG-metrics is also weak (cf. Table 2a).
We believe the reason for this imbalance of attribution quality may be due to the different magnitudes in the number of caption tokens and image patches. While captions typically have $\sim 10$ tokens, image representations in ViT-B-32 architectures consist of $\sim 200$ patches. Therefore, the ratio of the number of samples $N$ and tokens is much better than for image patches and the surrogate model $\varphi$ might be able estimate their importances better.

Overall, we find that the optimization of ILIME is quite sensitive to hyper-parameter choices and requires extensive tuning to find a setting that leads to stable convergence. In contrast, our method does not require additional optimization and involves no hyper-parameters except the number of integration steps $N$, whose increase must, however, improve attributions due to Equation 9.

# I    Implementation Details

For the implement of our method, we make use of the auto-differentiation framework in the PyTorch package. For a give input $\mathbf{x}(\alpha_n)$, $\mathbf{g}(\mathbf{x}(\alpha_n))$ is the forward pass through the encoder $\mathbf{g}$, and the Jacobian $\partial\mathbf{g}_k(\mathbf{x}(\alpha_n))/\partial\mathbf{x}_i$ is the corresponding backward pass. For an efficient computation of all $N$ interpolation steps in Eq. 9, we can batch forward and backward passes since individual interpolations are independent of another.
In practice, we attribute to intermediate representations, thus, the interpolations in Eq. 6 are between latent representations of the references and inputs. We use PyTorch *hooks* to compute these interpolations during the forward pass. Algorithm 1 sketches PyTorch-like pseudo-code of the implementation.

The application of our method to a different model or architecture only requires the implementation of a single forward hook. Registering hooks into models is a standard feature in auto-differentiation frameworks and does not require any modification of the given model's original code. The remaining steps to generate our attributions are differentiation through standard backpropagation and, finally, simple matrix multiplication to compute Eq. 10.

## J  Computational Complexity

Since the computation of the interpolated inputs $\mathbf{x}(\alpha_n)$ can be performed in parallel, $N$ is a constant with regard to time complexity. To build the full Jacobians of the encoders, however, we need to compute a separate backward pass for each output dimension, because auto-differentiation can only compute backward passes for scalar-valued outputs. Time complexity is dominated by this aspect and is thus on the order of $O(D)$, with $D$ being the embedding dimensionality of the output. The `intergrated_jacobian` method in Algorithm 1 sketches this computation.

Due to the fact that we typically attribute to intermediate representations, however, we do not need to compute full backward passes. Backpropagation can be stopped once it reaches the representation we attribute to, which results in this operation to be cheaper the deeper the representation of interest is. I.e. attributing to layer eleven is cheaper than attributing to layer five.

After building the Jacobians, the final attributions are computed through the matrix multiplications in Eq. 10, which is outlined at the bottom of Algorithm 1. Its computation time is negligible when calculated on GPU, but can substantially add to the total time when performed on CPU.

Space-wise, our method requires storing two Jacobians with the dimensions $D \times (D \times S)$, and $D \times (D \times H \times W)$, since input/intermediate representations are still sequential ($S$) on the text side and patch-based ($H \times W$) on the image side. Thus, memory consumption scales quadratically on the order of $O(D^2)$, and we require large VRAM to handle the computation efficiently on GPU.

In contrast, first-order methods only require backpropagation of a single scalar output value, i.e. the similarity score, whose result is a gradient vector as opposed to a Jacobian matrix. Hence, the cost of obtaining our second-order interaction attribution is the computation and handling of these Jacobians, which is substantially more expensive but enables a different level of insight into models that is not accessible through first-order methods.

**Algorithm 1** PyTorch-like pseudocode sketching the computation of our attributions. The syntax is simplified and not consistent. For a fully functional implementation, please refer to our GitHub repository. Comments in the pseudocode refer to the corresponding equations in Section 3.

```python
from torch import Tensor, arange, stack, autograd
import ExplainableCLIP, image_preparation, tokenize

model           = ExplainableCLIP(...)
image_input     = image_preparation(load_image("path/to/image.png"))
caption_input   = tokenize("some caption describing the image")

# Equations 6 and 7
def interpoloate(x: Tensor, ref: Tensor, n_steps: int):
    '''
    Compute n_steps linear interpolations between a reference ref and input x.
    '''
    step = 1 / n_steps
    alphas = arange(1, 0, step)  # interpolation coefficients
    x_interp = ref + alphas * (x - ref)  # interpolated representations
    return x_interp

# Equation 9
def integrated_jacobian(embedding: Tensor, intermediate: Tensor):
    '''
    Compute the integrated Jacobian for an embedding w.r.t. an
    intermediate representation.
    '''
    gradients = []
    for dim in range(embedding.size(0)):
        grad_d = autograd.grad(embedding[dim], intermediate)
        gradients.append(grad_d)
    jacobians = stack(gradients)
    # Integration over interpolations stacked along the first dimension
    int_jacobian = jacobians.sum(dim=0)
    return int_jacobian

# place hooks in the model to compute interpolations (not actual hook syntax)
image_hook   = model.register_hook(interpolate, image_layer, image_reference,   n_steps)
caption_hook = model.register_hook(interpolate, text_layer,  caption_reference, n_steps)

# Compute embeddings and retrieve intermediate representations from hooks
image_embedding   = model.encode_image(image_input)
caption_embedding = model.encode_caption(caption_input)
image_inter, image_ref_inter     = image_hook.get_intermediate_representation()
caption_inter, caption_ref_inter = caption_hook.get_intermediate_representation()

# Equation 10
image_jacobian   = integrated_jacobian(image_embedding,   image_inter)
caption_jacobian = integrated_jacobian(caption_embedding, caption_inter)
# Matrix multiplication
JJ = caption_jacobian.T @ image_jacobian
image_delta   = image_inter   - image_ref_inter
caption_delta = caption_inter - caption_ref_inter
# Element-wise multiplication with broadcasting
attributions = caption_delta * JJ * image_delta
```

