# OpenReview forum: "Explaining Caption-Image Interactions in CLIP Models with Second-Order Attributions"
_TMLR — Accepted by TMLR_

### Review · Reviewer_gQKD · 2025-04-08

**Summary Of Contributions:**

The authors note a research gap in attribution methods that are designed for non-symmetric dual-encoder models, such as CLIP.
They propose a novel second-order attribution method that can analyze interactions between modalities in dual encoders. Notably, their approach does not require any model fine-tuning or test-time optimization. They evaluate their method using CLIP on common benchmarks, ablations using input perturbations, and object localization.

**Audience:**

Yes

**Claims And Evidence:**

Yes

**Requested Changes:**

- Implement the proposed changes to Figure 1 or other ideas to make the Figure more readable
- Add high-level intuition to the method section to motivate the derivation (making the paper easier to follow)

I want to note that I am not well familiar with the related work in this area.

**Strengths And Weaknesses:**

**Strengths:**
- To the best of my knowledge the paper tackles an underexplored research domain
- Method design is principled (although difficult to read). The inter- and intra-model attribution method is flexible and effective
- Strong empirical results on different tasks.
- Convincing example of using the method to detect model failures (i.e., out-of-domain effects and subsequent qualitative failure analysis)

**Weaknesses:**
- I really like Figure 1. However, it is a bit confusing and can be slightly improved. It would be helpful to notate the individual images with a), b), etc. to make it easier to point the reader to the specific image in the caption. "Left" is ambiguous for two rows for example.
- The method section lacks intuition and is really difficult to read. No single step is motivated, which makes it hard to understand why certain things are introduced in the first place. Here, some high-level intuition would make the paper easier to read. The end of the first paragraph contains some intuition. The authors could try to state the goal of their derivation at the beginning of the section.

---

> ### Author Response · Authors · 2025-04-18
> **Response to review**
>
> Thank you for the positive feedback!
>
> We agree that Figure 1 could be better structured and will work on improving it.
> It is important for us that this figure is easy to understand because our goal is to provide an intuitive understanding of our method here, especially pointing out how it is different from first-order attributions. Therefore, we appreciate all specific feedback we can get about it.
>
> We also acknowledge that the theoretical part of our work in Section 3 is not very intuitive.
> We would like this to be as accessible as possible. However, on the other hand, it is a formal derivation and we would also like to keep the formality in this section.
> As a compromise, we will especially try to do a better job motivating the starting point in the beginning and add a more intuitive explanation of how to interpret the result.
>
> We will address both aspects in the revised version.

---

> ### Author Response · Authors · 2025-04-18
> **Intuition behind the method**
>
> Below is a walk through the derivation in Section 3 with an attempt to add intuitive illustration wherever it is possible.
> We will try to add parts of it to the paper where it appears appropriate.
>
> First, Eq. 1 formally defines the dual encoder architecture as a function $f$ computing the dot-product between two embeddings for the inputs $a$ and $b$.
> We then define two special inputs $r_a$ and $r_b$, for which we require that they must be dissimilar to any other input, but this will only become important later.
> Now comes the most rigorous point of the derivation. With these four inputs, we simply write down Eq. 2. At this point there is no intuition behind this.
> However, as mentioned between Eq. 2 and 3, we can show the equality of this equation to Eq. 10 and then reduce it to our final attributions in Eq. 11. For now, we will not care about why this equality is useful after all. We will first focus on deriving it step by step.
>
> The first step in Eq. 3 is to reformulate Eq. 2 as an integral over the second-order derivative of $f$. This step will be a lot more intuitive when reading it backwards. In reverse order it is just standard integration first plugging in the bounds of the inner integral over $x$ and then repeating the procedure for the outer integral over $y$.
> We plug in the model definition from Eq. 1 to replace $f$ by the product of the two embeddings to get to Eq. 4. We then see that all derivatives and integrals can be separated because encoder $g$ does not depend on input $y$ and vice versa. This is just a product rule.
> The result in Eq. 5 is a product of two integrals over the Jacobians of the two encoders $g$ and $h$. So in the following, we can treat them separately and plug the result back into Eq. 5 when we are done, which comes later though.
>
> The two integrals are in high-dimensional spaces, because the inputs of the encoders (and the bounds of the integrals) are feature representations of texts and images. Therefore, they are line integrals and we can imagine them as walking from the lower bound (the reference $r_a$ or $r_b$) to the upper bound (our actual inputs $a$ and $b$) in these feature spaces. On the way, we collect all the Jacobians and integrate over them. To do this we need to define a path from the lower to the upper bound that we walk along. We simply use straight lines here, that are defined in Eq. 6 and 7.
> To integrate along these lines, we substitute them into the integrals as shown for encoder $g$ in Eq. 8. This changes the integral over high dimensional coordinates to an integral over the scalar $\alpha$. As we increase $\alpha$, we proceed walking from $r_a$ to $a$ on the straight line between them. The wikipedia article about line integrals has some very nice and intuitive animations visualizing this (https://en.wikipedia.org/wiki/Line_integral).
> Note, that this is also exactly what integrated gradients do for scalar-valued models (e.g. classification or regression models). The only difference is that here we deal with encoder models producing vector representations of inputs. Therefore, we get Jacobians instead of gradients.
>
> To proceed with Eq. 8, we need to calculate the derivative of the integration path $x(\alpha)$ w.r.t $\alpha$, which is constant (since it’s linear). Hence, we can pull it out of the integral. This leaves us with an integral over the Jacobian of the encoder with the scalar integration variable $\alpha$. In Eq. 9 we put a name to these integrals and call them “integrated Jacobians”, denoted by $J$.
> To calculate them in practice, we take $N$ discrete steps along our integration path, i.e. N values for $\alpha$ between zero and one and approximate the integral by a sum over these steps. This is just numerical integration.
>
> Finally, we do plug everything back into Eq. 5 again as mentioned above and arrive at Eq. 10.
> At this point it is worth talking about the indices over features $i$ in input $a$ and features $j$ in input $b$. The expression in Eq. 5 contains both. Therefore, we can think of it as a matrix with indices $i$ and $j$. We will pick this up again further down. For now, we just call this matrix $A_{ij}$. Before we get to interpreting it let us zoom out because now we have shown the equality between Eq. 2 and Eq. 10 that we mentioned in the beginning. In short, we now have:
>
> $f(a, b) - f(a, r_b) - f(r_a, b) + f(r_a, r_b) = \sum_{ij} A_{ij}$
>
> Note, that here we explicitly write out the sum over the indices $i$ and $j$. Throughout the derivation, we used the convention that double indices are summed over for an uncluttered notation (also ‘Einstein sum notation / convention’).

---

> ### Author Response · Authors · 2025-04-18
> **continuation**
>
> This is now where the references $r_a$ and $r_b$ become important again. Earlier, we defined them to be uninformative, i.e. leading to a similarity score of $s\approx 0$ when plugged into Eq. 1, so that $f(a, r_b) \approx f(r_a, b) \approx f(r_a, r_b) \approx 0$.
> If this is the case, we can approximately cancel them out in the equation above which then yields the final result in Eq. 11:
>
> $f(a, b) \approx \sum_{ij} A_{ij}$
>
> This is the end of the derivation. Let’s take a moment to interpret this result now. On the left side we have the model prediction itself, which is a scalar similarity score. On the right side, we have a matrix of terms, each involving one feature from the first and one from the second input. Thus, it describes the contribution of all possible interactions between pairs of features in the two inputs. If we sum over all entries in this matrix the result must approximately equal the predicted similarity score. Therefore, we can think of this matrix as an approximate breakdown of the model prediction into additive contributions of individual feature interactions between the two inputs.
> Since the image features are two-dimensional and the text sequence is one-dimensional, this interaction matrix has three dimensions. To visualize it we, require the slicing that is described in the following section and that we demonstrate in Figure 1.

---

> > ### Comment · Reviewer_gQKD · 2025-04-18
> > **Feedback**
> >
> > Thanks for the response.
> >
> > The changes you propose to the method section are exactly what I had in mind. I think it will make the paper slightly easier to read.
> > I am looking forward to changes to Figure one (but it is a relatively minor concern)

---

> > > ### Author Response · Authors · 2025-05-19
> > > **Revision**
> > >
> > > We have uploaded a revised version of our manuscript including an improved structure for Figure 1. Figure 22 in the Appendix also includes an alternative visualization. We would appreciate your feedback on which is more easily understandable.

---

### Review · Reviewer_zVe1 · 2025-04-10

**Summary Of Contributions:**

This paper focuses on performing feature attribution on dual-encoder based models to improve their overall explainability. The authors address this by deriving a decomposition of a model prediction into additive contributions from features for each individual modality (Eq 11). When averaging over feature dimensions, this produces a 3D attribution tensor. The authors compare this approach against other feature attribution baselines that only look at first-order interactions.

The authors provide evaluations for CLIP models on a variety of different tasks, including attribution evaluation and model analysis. Their attribution evaluation setting is comprised of multiple tasks: (1) perturbing most important features and evaluating the resulting change in similarity scores; (2) visualization object localizations of texts mentioned in the caption – both qualitatively and quantitatively through the Point Game framework. Their model analysis evaluation is comprised of measuring OOD generalization, class-wise grounding, and object localization.

Their empirical results show that (1) CLIP models show more fine-grained interactions between the inputs of two different modalities and (2) CLIP models struggle to differentiate between objects appearing together and performs poorly out of distribution. The authors also show that (2) can be addressed through domain-specific finetuning.

**Audience:**

Yes

**Claims And Evidence:**

Yes

**Requested Changes:**

Some critical points to address:
* The authors should explore more about the approximate error in the attribution method, specifically discussing how the choice of reference inputs and the number of integration steps affect the accuracy of attributions
* The authors should include a discussion on the computational overhead of the method compared to first-order attribution techniques, as this is a natural concern when increasing the complexity of measured feature interactions

Recommended change to strengthen the work:
* One of my main concerns is the practicality or pragmatic uses of this evaluation. Wat are some of the use cases of these new insights that have been gained? How do they inform how CLIP models should be trained?
* Added discussion of related works mentioned above.

**Strengths And Weaknesses:**

## Strengths

A derivation of a new second-order feature attribution approach that is principally grounded (although remaining an approximation).

The authors apply this technique to highlight existing failure modes of CLIP models.

## Weaknesses

There’s somewhat of a lack of how these empirical findings can be leveraged to improve dual encoder architectures (and their training) and their downstream usage.

There’s no real discussion on the impact of this approximation — how good / bad is this approximation in practice?

Lack of discussion with other works that seem to make quite similar findings of issues in CLIP models [1, 2]



[1] Lewis, et. al. Does CLIP Bind Concepts? Probing Compositionality in Large Image Models

[2] Sam, et. al. Finetuning CLIP to Reason about Pairwise Differences.

---

> ### Author Response · Authors · 2025-04-24
> **Response to review**
>
> Thank you for your constructive feedback!
>
> We agree that explicitly evaluating the effect of the number of integration steps ($N$) and different references would be interesting. We have actually run initial experiments in this direction but decided to focus on other experiments to keep to the page limit.
> However, we have now picked up this work again, and have run a systematic evaluation.
> The Table below shows results on the approximation error (i.e. the absolute difference between Eq. 1 and Eq. 10) as a function of $N$. As expected, for higher $N$, the error converges.
>
> | N | approximation Error |
> |-|-|
> | 1 | $0.149 \pm 0.048$ |
> | 5 | $0.080 \pm 0.021$ |
> | 10 | $0.064 \pm 0.019$ |
> | 50 | $0.051 \pm 0.017$ |
> | 100 | $0.049 \pm 0.017$ |
>
> We have also repeated this experiment for different references. On the image side, next to the black image, we have added an image of zero-centered gaussian noise. On the text side, next to a sequence of padding tokens, we added the empty sequence consisting only of the CLS and EOS tokens. All combinations of references show good convergence of the approximation error, with some converging slightly faster than others.
> In the revised version of the paper, we will add a section for these results in the experiments including plots of the convergence behavior.
>
> Regarding the additional points to discuss:
> - We will include the two cited references. Thank you for pointing them out.
> - Appendix F provides details about the implementation of our method and also touches some aspects that are relevant for its computational overhead. However, we agree that computational costs are not discussed explicitly and we will include more details in the revised version.
> - The first paragraph of our Discussion (Section 5) also addresses what our findings, especially the five error categories, may imply for training contrastive image-caption models in the future. With more space for a final version, we can certainly extend this discussion. However, we also want to emphasize the scope of this work. Our main contribution is a new second-order attribution method, its evaluation and use as a diagnostic tool for already trained models. Utilizing the gained insights to improve the models will be interesting future work but exceeds the scope of this contribution.

---

> > ### Author Response · Authors · 2025-05-19
> > **Revision**
> >
> > We have uploaded a revision of our manuscript including an additional experiment on our attributions' approximation error as well as different references in Appendix C and a discussion of computational costs in Appendix I among other changes outlined in our above response to all. We will appreciate your feedback on these changes.

---

### Review · Reviewer_6F2A · 2025-05-10

**Summary Of Contributions:**

This paper introduces a new attribution method develop to explain the functioning of dual-encoder models like CLIP by computing pair-wise attributions between cross-modal features (like text and images). The method is derived as a second-order method, but the factorized formulation of dual-encoders allows the authors to obtain an expression that is relatively simple and computationally manageable, and results formally very similar to the Integrated Gradients method (in practice, the expression of the final cross-modal attributions is essentially the product of Integrated Gradients).

In the case of CLIP, the final attribution map is a 3D tensor of size caption sequence length x image height x image width, which sliced in the text sequence length dimensions provides a 2D attribution map indicating the pixel contributions to explaining the text tokens in the slice, and sliced in the image height and width dimensions provides a 1D attribution map indicating the text token contributions.
The method is extensively empirically evaluated by applying it on OpenAI's CLIP model, as well as the open models MetaCLIP and OpenCLIP, and compared against a host of alternative baseline explainability methods on multiple experiments to quantify faithfulness of the proposed method through patch deletion, correspondence of the localization of the attribution maps with human bounding-box annotation, and testing the effect of hard negative captions.
In most experiments, the proposed methods outperforms the baselines.

Finally, the paper showcases the use of the method to diagnose the functioning of CLIP-like models, by evaluating the effect of fine-tuning to mitigate out-of-domain shifts, and using it to uncover and classify failure modes which presumably diagnose issues in how the CLIP model is trained.

**Audience:**

Yes

**Claims And Evidence:**

Yes

**Requested Changes:**

* Since the feature-pair interaction sum to f(a,b), the total attributions for one modality (obtained by marginalizing pairwise interactions over the other) should sum to 0 (under the assumption of uninformative references used in the Methods section). Could you discuss the implications of this? For instance, it seems that for each group of positive pixels, there will have to be a corresponding group of negative pixels, which seems like a constraint of sort, since one could imagine an image consisting exclusively of positive pixels and an uninformative background. However, the proposed method does not contemplate this scenario. Could you please elaborate on this? The paragraph 'Object Discrimination' notices this phenomenon, but discusses it as seemingly accidental and diagnostic of the model's functioning, while in fact it presumably should be discussed as an inherent property of the attribution method. If this is indeed the case, then even some of the main conclusions of the paper should be rephrased to reflect the fact that for instance the uncovered penalization of mismatches by assigning negative attributions discussed in the Discussion and Conclusion sections might not indeed necessarily be a property of CLIP models, but a phenomenon of the attribution method.
* Provide a discussion on the computational complexity of the method, and consider adding pseudocode to show in practice how attributions are computed.
* Discuss the choice of reference inputs and the choice in integrating along a linear path between references and inputs (eq. 6 and 7).
* Explain what uncertainty intervals mean, e.g. in Fig. 3.

**Strengths And Weaknesses:**

### Strengths
- The method's derivation is clear and explained straightforwardly and the resulting cross-modal attributions are intuitive
- The methods is validated quite extensively, and the authors provide a large number of experiments to show the method's performance in terms of faithfulness of the provided attributions compared to the main alternatives
- The method is computationally parsimonious, making for a practical attribution algorithm
- The method is used to provide convincing insights on the effects of out-of-domain deployment vs fine-tuning

### Weaknesses
- Relation to Integrated Gradients is not sufficiently discussed
- The computational complexity of the method scales with H x W x S x d (the encoder output dimension) x N (the integration steps) x the cost of computing a gradient through the encoder. This might be efficient, as the paper indicates, but it might still be considerable, especially for large images and long captions. The paper might benefit from a more detailed discussion on the computational cost of running the argument.

---

> ### Author Response · Authors · 2025-05-15
> **Response to review**
>
> Thank you for taking the time to review our work.
>
> From your suggestions, we identify the following central points that we will address in the revised version:
> - We will elaborate more on computational complexity as already discussed with reviewer zVe1. Pseudo-code will be a good addition.
> - We can certainly point out the relation to integrated gradients more, regarding both the derivation and the final results. The choice of integrating along a straight line is closely related to this aspect since the original paper also parameterizes the integration path linearly. A step that makes explicit use of the linearity of the path is Equation 8. For a non-linear parameterization, we could not move the derivation $\partial x(\alpha) / \partial \alpha$ out of the integral, which would yield a slightly different theoretical result. It would be interesting future work to evaluate whether and how different non-linear parameterizations affect attributions. At the same time, we argue that in the absence of evidence for the need of a more complex parameterization, the canonical linear one should be the first choice.
> - The choice of references is also an interesting aspect that was brought up by reviewer zVe1 as well. We have, by now run an experiment implementing different references. On the image side we additionally implement gaussian noise next to the black image and on the text side we include an empty sequence, next to padding tokens. The revised version includes an evaluation of the attribution error, i.e. the absolute difference between Eq. 2 and 11 for different combinations of references and also shows examples.
> - We will also include a note on the shown uncertainty intervals. They are standard deviations over respective validation splits.
>
> Regarding your comment on marginalization, we would like to clarify a few things.
> You mentioned that the ‘total attribution to one modality obtained by marginalizing over the other should sum to 0’.
> This is actually not the case. When marginalizing over one mode, the result is an attribution map over the other mode that still sums to the predicted similarity score, not to zero. If we e.g. sum over the caption, we get an attribution map over the image representing interactions between individual image patches and the full caption. In contrast, the raw attributions are for interactions between image patches and individual tokens in the caption.
> Essentially, we are free to combine arbitrary selections over attributions by summation (due to the sum in Eq. 11) including marginalizing over one input mode entirely. But the total of all attributions must always equal Eq. 2 and approximate the similarity score $f(a, b)$ (up to the numerical integration error). We can never “lose” attributions through summation generally and marginalization specifically.
> Does this address your concern?

---

> > ### Author Response · Authors · 2025-05-19
> > **Revision**
> >
> > We have uploaded a revised version of our manuscript including an experiment on the choice of references in Appendix C and a discussion of computational costs in Appendix I with supporting pseudocode in Algorithm 1. In the introduction, methods and discussion sections, we also discuss the relation to integrated gradients and the choice of a linear integration path more elaborately. We will appreciate your feedback on these changes.

---

> ### Comment · Reviewer_6F2A · 2025-05-26
>
> Thank you very much the clarifications, in particular regarding the fact that summing the attribution maps will sum to the predicted similarity score. Of course, that makes sense. I was thrown off by the assumption of uninformative references for which similarity is zero.
> I also appreciate the additions in the new version of the manuscript, including the clarification about memory and computational complexity and the expanded discussions relating the work to integrated gradients which address the comments that I had provided in my review.

---

> > ### Author Response · Authors · 2025-05-27
> > **acknowledgement**
> >
> > Thank you for getting back to us. We are glad we could clarify the aspect regarding summation of attributions and address your comments in the revision.

---

### Author Response · Authors · 2025-05-19
**Revision**

Dear reviewers and editors,

We have uploaded a revised version of our work addressing suggestions and requests in the reviews. Specifically:
- We have added an experiment evaluating our method’s approximation error as a function of the number of integration steps $N$ in Appendix C.
- This experiment also includes different references for both the image and text encoder and evaluates differences in the resulting attributions.
- We have added a discussion of computational complexity in Appendix I including pseudocode in Algorithm 1.
- We have improved the structure of Figure 1. Figure 22 in the Appendix also shows an alternative illustration and we will appreciate feedback on which version is more intuitive and understandable.
- We have changed parts of the introduction, methods and discussion sections to elaborate more on the relation to integrated gradients, the linearity of the integration path and to improve the intuitive understanding of our method.
- We have added the suggested related work.

---

### Decision · Action_Editor_h6j5 · 2025-06-26

**Recommendation:** Accept as is

**Audience:**

Yes

**Audience Explanation:**

Attribution for explainability is of general interest to the community, particularly given that this applied to a model that has served as a core workhorse for many. Ideally this leads to a practical tool for practitioners understanding model failures.

**Claims And Evidence:**

Yes

**Claims Explanation:**

The reviewers feel the claims are extensively validated, the idea interesting and theoretically sound, and the challenge underexplored. The approach performs well against baselines. The key concerns or questions were addressed in the revised submission, including several new additions (experiments and pseudocode) to the appendix in addition to improvements of clarity.  The result is that all reviewers are positive on the current manuscript.